# OPTIMIZING ADAPTIVE ATTACKS AGAINST CONTENT WATERMARKS FOR LANGUAGE MODELS

## ABSTRACT

Large Language Models (LLMs) can be *misused* to spread online spam and misinformation. Content watermarking deters misuse by hiding a message in model-generated outputs, enabling their detection using a secret watermarking key. Robustness is a core security property, stating that evading detection requires (significant) degradation of the content's quality. Many LLM watermarking methods have been proposed, but robustness is tested only against *non-adaptive* attackers who lack knowledge of the watermarking method and can find only suboptimal attacks. We formulate the robustness of LLM watermarking as an objective function and use preference-based optimization to tune *adaptive* attacks against the specific watermarking method. Our evaluation shows that (i) adaptive attacks substantially outperform non-adaptive baselines. (ii) Even in a non-adaptive setting, adaptive attacks optimized against a few known watermarks remain highly effective when tested against other unseen watermarks, and (iii) optimization-based attacks are practical and need limited computational resources of less than seven GPU hours. Our findings underscore the need to test robustness against adaptive attackers.

## 1 INTRODUCTION

Few providers of Large Language Models (LLMs) empower millions of users to generate human-quality text at scale, raising concerns about dual use (Barrett et al., 2023). Untrustworthy users can *misuse* the provided LLMs to generate harmful content, such as online spam (Weidinger et al., 2021), misinformation (Chen & Shu, 2024), or to facilitate social engineering attacks (Shoaib et al., 2023). The ability to detect model-generated text can control these risks (Grinbaum & Adomaitis, 2022).

Content watermarking enables the detection of model-generated outputs by embedding hidden marks that can be extracted with a secret watermarking key. Some LLM providers, such as DeepMind (DeepMind, 2024) and Meta (San Roman et al., 2024), have already deployed watermarking to promote the ethical use of their models. A threat to these providers are users who perturb model-generated text to evade watermark detection while preserving text quality. Such undetectable, model-generated text could further erode trust in the authenticity of digital media (Federal Register, 2023).

A core security property of watermarking is *robustness*, which requires that evading detection is only possible by significantly degrading text quality. Testing robustness requires identifying the most effective attack against a specific watermarking method. However, existing content watermarks for LLMs (Kirchenbauer et al., 2023a; Aaronson & Kirchner, 2023; Christ et al., 2023; Kuditipudi et al., 2024) test robustness only against *non-adaptive* attackers, who lack knowledge of the watermarking algorithms. This reliance on obscurity makes watermarking vulnerable to *adaptive* attacks (Lukas et al., 2024; Nicks et al., 2024) when information about the watermarking algorithms is leaked.

We propose a method to curate preference datasets and adaptively optimize an attack against *known* content watermarking algorithms. Optimization is challenging due to (i) the complexity of optimizing over the discrete textual domain and (ii) the limited computational resources available to attackers. We demonstrate that adaptively tuned, open-weight LLMs such as Llama2-7b (Touvron et al., 2023) evade detection at negligible impact on text quality against Llama3.1-70b (Dubey et al., 2024). Our attacker spends less than 7 GPU hours to achieve over 96% evasion rate against any of the surveyed watermarking methods with negligible impact on text quality. Our attacks are Pareto optimal, even in the non-adaptive setting where they transfer to unseen watermarks. Hence, future watermarking methods must consider our attacks to test robustness.

## 1.1 CONTRIBUTIONS

We make the following contributions.

- We propose methods to curate preference-based datasets using LLMs, enabling us to adaptively fine-tune watermark evasion attacks against four state-of-the-art language watermarks.
- Adaptively tuned paraphrasers with 0.5-7 billion parameters evade detection from all tested watermarks at a negligible impact on text quality. We demonstrate their Pareto optimality for evasion rates greater than 90%[1]. Optimization requires less than seven GPU hours.
- We test our optimized attacks in the non-adaptive setting against unseen watermarks and demonstrate that they remain Pareto optimal compared to other non-adaptive attacks. Our results underscore the necessity of using optimized, adaptive attacks to test robustness.
- We will release our source code and adaptively tuned paraphrasers to facilitate further research on robustness against adaptive attackers.

## 2 BACKGROUND

**Large Language Models (LLMs)** estimate the probability distribution of the next token over a vocabulary $\mathcal{V}$ given a sequence of tokens. Autoregressive LLMs predict each subsequent token based on all preceding tokens. Formally, for a sequence of tokens $x_1, \ldots, x_n$, an LLM models:

$$P(x_n | x_1, \ldots, x_{n-1}) = \text{softmax}(f_\theta(x_1, \ldots, x_{n-1}))_n$$

where $f_\theta$ is a neural network with parameters $\theta$. Optimizing LLMs to maximize a reward function is challenging because the text is discrete, and the autoregressive generation process prevents direct backpropagation through the token sampling steps (Schulman et al., 2017).

**LLM Content Watermarking** hides a message in model-generated content, such as text, that can later be extracted with access to the content using a secret watermarking key. A *watermarking method* is a set of algorithms (KEYGEN, EMBED, VERIFY) formalized as follows (Lukas et al., 2024).

- $\tau \leftarrow \text{KEYGEN}(\theta, \gamma)$: A randomized function to generate a watermarking key $\tau$ given secret (i) LLM parameters $\theta$ and (ii) random seeds $\gamma \in \mathbb{R}$.
- $\theta^* \leftarrow \text{EMBED}(\theta, \tau, m)$: Given a LLM $\theta$, a watermarking key $\tau$ and a message $m$, this function returns parameters $\theta^*$ of a *watermarked* LLM that generates watermarked text.
- $\eta \leftarrow \text{VERIFY}(x, \tau, m)$: Detection requires (i) extracting a message $m'$ from text $x$ using $\tau$ and (ii) returning the $p$-value $\eta$ to reject the null hypothesis that $m$ and $m'$ match by chance.

A text watermark is a hidden signal in text that can be mapped to a message $m \in \mathcal{M}$ using a secret watermarking key $\tau$. The key $\tau$ refers to secret random bits of information used to detect a watermark. A watermark is *retained* if the verification procedure returns $\eta < \rho$, for $\rho \in \mathbb{R}^+$. Let $Q : \mathcal{V}^* \times \mathcal{V}^* \rightarrow \mathbb{R}$ be a function to measure the quality between pairs of texts. For model-generated text $x$, the watermarking method is *robust* if an attacker can only generate a paraphrased text $x'$ with text quality greater than $\delta \in \mathbb{R}$ that does not retain the watermark with probability at most $\epsilon \in \mathbb{R}^+$.

$$\Pr\left[\text{VERIFY}(x', \tau, m) \geq \rho \ \wedge \ Q(x, x') > \delta\right] < \epsilon \tag{1}$$

**Evasion Attacks.** Watermark evasion attacks are classified by the attacker's access to the provider's (i) LLM, (ii) detection algorithm VERIFY that uses the provider's secret watermarking key, and (iii) knowledge of the watermarking algorithms. A *no-box* attacker has no access to the provider's LLM, whereas *black-box* attackers have API access, and *white-box* attackers know the parameters of the provider's LLM. *Online* attackers can query the provider's VERIFY functionality, as opposed to *offline* attackers who have no such access. *Adaptive* attackers know the algorithmic descriptions (KEYGEN, EMBED, VERIFY) of the provider's watermarking method, while *non-adaptive* attackers lack this knowledge. Our work focuses on no-box, offline attacks in adaptive and non-adaptive settings.

---

[1] Closed models such as GPT4o are on the Pareto front due to high text quality but have lower evasion rates.

**Surveyed Watermarking Methods.** Following Piet et al. (2023), we evaluate the robustness of four state-of-the-art watermarking methods. The `Exp` (Aaronson & Kirchner, 2023) method marks text by selecting tokens that maximize a score combining the conditional probability $P(x_n \mid x_0 \ldots x_{n-1})$ and a pseudorandom value derived from a sliding window of prior tokens. The `Dist-Shift` (Kirchenbauer et al., 2023a) method favours tokens from a green list, which is generated based on pseudorandom values and biases their logits to increase their selection probability. The `Binary` (Christ et al., 2023) approach converts tokens into bit-strings determined by pseudorandom values and the language model's bit distribution, subsequently translating the bit-string back into a token sequence. Lastly, the `Inverse` (Kuditipudi et al., 2024) scheme uses inverse transform sampling by computing a cumulative distribution function ordered pseudorandomly according to a secret key and using a fixed pseudorandom value to sample from this distribution. We refer to Piet et al. (2023) for more details.

## 3 THREAT MODEL

We consider a provider capable of training LLMs and deploying them to many users via a black-box API, such as Google with Gemini or OpenAI with ChatGPT. The threat to the provider are untrustworthy users who misuse the provided LLM and generate harmful content without detection.

**Provider's Capabilities and Goals** *(Deployment)* The provider fully controls the LLM and the text generation process, including the ability to embed a watermark into model-generated text. *(Watermark Verification)* The provider must be able to verify their content watermark in each model-generated text. Their goal is to have an (i) quality-preserving and (ii) robust watermark that enables detection of model-generated text at a given, low False Positive Rate (FPR) $\rho \in \mathbb{R}^+$.

**Attacker's Capabilities.** *(Access Restrictions)* We consider a (i) no-box attacker who cannot collect watermarked texts during training but is (ii) offline, meaning that they cannot access the provider's VERIFY function. Our focus is on (iii) adaptive attackers, who know the provider's watermark method (KEYGEN, EMBED, VERIFY) but do not know the secret inputs used for watermarking, such as the provider's LLM or random seeds. We also evaluate how adaptive attacks transfer to the non-adaptive setting against unseen watermarks. *(Surrogate Models)* A surrogate model is a model trained for the same task as the provider's model. For example, while ChatGPT3.5's weights are not public, the attacker could access the parameters of smaller, publicly released models such as those from the `Llama2` (Touvron et al., 2023) model family from open model sharing platforms. Our attacker can access such open-weight *surrogate* models and use them for *paraphrasing* text. We assume the surrogate model's text quality is inferior to the provided model; otherwise, there would be no need to use the watermarked model. **Attacker's Goals.** The attacker wants to use the provided, watermarked LLM to generate text (i) without a watermark that (ii) has a high quality. We measure quality using many metrics, including a quality function $Q : \mathcal{V}^* \times \mathcal{V}^* \to \mathbb{R}$ between pairs of text when the attacker attempts to evade detection. We require that the provider correctly verifies their watermark with a given p-value threshold $\eta < \rho$ for $\rho \in \{0.01, 0.025, 0.05, 0.075, 0.1\}$. Lower p-values make evasion more likely to succeed for the attacker (i.e., detection becomes more challenging for the provider).

## 4 CONCEPTUAL APPROACH

We adaptively fine-tune an open-weight paraphraser $\theta_P$ against given watermarking methods. The attacker lacks knowledge of the provider's watermarking key $\tau \leftarrow$ KEYGEN$(\theta, \gamma)$, which depends on the (unknown) parameters $\theta$ of the provider's LLM and random seed $\gamma$. Our attacker overcomes this uncertainty by choosing an open-weight surrogate model $\theta_S$ to generate so-called *surrogate* watermarking keys $\tau'$ and optimizes the expected evasion rate over many random seeds $\gamma \sim \mathbb{R}$.

### 4.1 ROBUSTNESS AS AN OBJECTIVE FUNCTION

Let $P_\theta : \mathcal{V}^* \to \mathcal{V}^*$ denote a paraphrasing function, $H_\theta : \mathcal{V}^* \to \mathcal{V}^*$ is a function to produce model-generated text given a query $q \in \mathcal{V}^*$ and $Q : \mathcal{V}^* \times \mathcal{V}^* \to \mathbb{R}$ measures the similarity between pairs of text. We formulate robustness using the following objective function that we can optimize.

$$\max_{\theta_P} \mathbb{E}_{\substack{\gamma \sim \mathcal{R} \\ m' \sim \mathcal{M} \\ q \sim \mathcal{T}}} \mathbb{E}_{\substack{\tau' \leftarrow \text{KeyGen}(\theta_S, \gamma) \\ \theta_S^* \leftarrow \text{Embed}(\theta_S, \tau', m') \\ r \leftarrow H_{\theta_S^*}(q)}} \left[ \text{Verify}\left(P_{\theta_P}(r), \tau', m'\right) + Q\left(P_{\theta_P}(r), r\right) \right] \qquad (2)$$

Equation (2) finds optimal parameters for the paraphraser $\theta_P$ by sampling uniformly at random over (i) random seeds $\gamma \sim \mathbb{R}$, (ii) messages $m' \sim \mathcal{M}$ and (iii) queries $q \sim \mathcal{T}$. The second expectation is taken over a surrogate watermarking key, using the surrogate model's parameters $\theta_S$ and the previously sampled random seed $\gamma$ as input. The surrogate model, key and message are used to embed a watermark into the surrogate model $\theta_S^*$ to generate a watermarked sample $r$. The optimization aims to find optimal parameters $\theta_P^*$, so the paraphraser has a high probability of generating text $r' \leftarrow P_{\theta_P}(r)$ that evades watermark detection and preserves text quality compared to $r$.

Optimization presents multiple challenges. The attacker optimizes over different random seeds $\gamma$ and a surrogate model $\theta_S$ instead of the provider's model $\theta$, which may affect the attack's effectiveness when testing against the provider's watermark. The discrete nature of text and the inability to backpropagate through the text generation process make maximising the reward challenging. Furthermore, the reward function depends on Verify, which may not be differentiable. Deep reinforcement learning (RL) methods (Schulman et al., 2017; Rafailov et al., 2024) do not require differentiable reward functions. However, RL is known to be compute-intensive and unstable, making it unclear whether optimization can achieve a high reward using limited computational resources.

### 4.2 Preference Dataset Curation

We use existing reinforcement learning (RL) methods such as Direct Preference Optimization (DPO) (Rafailov et al., 2024) to optimize Equation (2). To optimize, DPO requires collecting a dataset of *positive* and *negative* examples and fine-tunes the paraphraser to increase the probability of generating positive examples while reducing the probability of generating negative examples. A *negative* sample retains the watermark and represents a failed attempt at watermark evasion. In contrast, positive samples do not retain a watermark and have a high text quality $Q(r, r'_p) > \delta$ for an attacker-chosen $\delta \in \mathbb{R}^+$. To bootstrap optimization, we require the ability to curate positive and negative examples. We bootstrap using publicly available, open-weight paraphrasers and best-of-N rejection sampling to curate triplets $(r, r'_n, r'_p)$ containing a watermarked sample $r$ and two paraphrased versions $r'_n, r'_p$ representing the negative and positive examples, respectively.

Algorithm 1 randomly samples from a set of known watermarking methods $\mathcal{W}$ (line 2) and from the set of task-specific queries $\mathcal{T}$ (line 3). It samples a message $m$ (line 4) and generates a surrogate watermarking key $\tau'$ to embed a watermark into the surrogate generator (lines 4-5). We generate text $r$ using the watermarked model $\theta_S^*$ (line 7) and verify whether it retains the watermark (line 8). The paraphrase model $\theta_P$ generates $N$ paraphrased versions of $r$ that we partition into positive and negative samples (lines 10-11). A sample $r'_p$ is positive ($b = 1$) if it does not retain the watermark and has high text quality $\geq \delta$ and negative $r'_n$ ($b = 0$) otherwise. For each positive sample, we select one corresponding negative sample and add the watermarked text and the negative and positive paraphrases to the preference dataset $\mathcal{D}$ (lines 15-17).

| Attack Name | Description |
|---|---|
| DIPPER (Krishna et al., 2023) | Train an 11b Sequence-to-Sequence model for paraphrasing. |
| Translate (Piet et al., 2023) | Translate to another language and back (e.g., French, Russian). |
| Swap (Piet et al., 2023) | Randomly remove, add or swap words. |
| Synonym (Piet et al., 2023) | Replace words with a synonym using WordNet (Miller, 1995). |
| HELM (Bommasani et al., 2023) | Randomly add typos, lowercase or contractions |
| Llama, Qwen, GPT3.5 | Paraphrase text using a publicly accessible LLM. |
| Ours-`Llama2-7b`-Exp | Paraphrase with a `Llama2-7b` model tuned adaptively against Exp. |

Table 1: (Top) The non-adaptive baseline attacks we consider in our study against (Bottom) our adaptively fine-tuned attacks. We refer to Piet et al. (2023) for details on the baseline attacks and Appendix A.7 for our adaptive attack.

**Algorithm 1** Preference Dataset Curation

**Require:** Surrogate $\theta_S$, Paraphraser $\theta_P$, Queries $\mathcal{T}$, Messages $\mathcal{M}$, Paraphrase Repetition Rate $N$, False Positive Rate Threshold $\rho$, Quality Threshold $\delta$

1: $\mathcal{D} \leftarrow \{\}$      ▷ The preference dataset
2: **for** (KEYGEN, EMBED, VERIFY) $\in \mathcal{W}$ **do**      ▷ Optimize over known watermarking methods
3:      **for** each query $q \in \mathcal{T}$ **do**
4:          $m \sim \mathcal{M}$
5:          $\tau' \leftarrow$ KEYGEN$(\theta_S, \text{RND}())$      ▷ Generate a surrogate watermarking key
6:          $\theta_S^* \leftarrow$ EMBED$(\theta_S, \tau', m)$      ▷ Watermark the surrogate model
7:          $r \leftarrow S_{\theta_S^*}(q)$      ▷ Generate watermarked text
8:          **if** VERIFY$(r, \tau', m) < \rho$ **then**
9:             $R^0, R^1 \leftarrow \{\}, \{\}$
10:             **for** i $\in$ [N] **do**      ▷ Best-of-N paraphrasing
11:                 $r' \leftarrow P_{\theta_P}(r)$      ▷ Generate a paraphrased sample
12:                 $b \leftarrow \begin{cases} 1 & \text{if VERIFY}(r', \tau', m) > \rho \wedge Q(r, r') \geq \delta, \\ 0 & \text{otherwise.} \end{cases}$
13:                 $R^b \leftarrow R^b \cup \{r'\}$
14:             **for** $j \in [|R^1|]$ **do**
15:                 $r_n' \leftarrow \begin{cases} R_j^0 & \text{if } |R^0| \geq j, \\ r & \text{otherwise.} \end{cases}$      ▷ Choose a negative sample
16:                 $\mathcal{D} \leftarrow \mathcal{D} \cup \{(r, r_n', R_j^1)\}$
17: **return** $D$

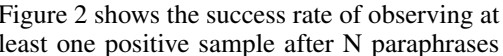

Figure 1: An algorithm to curate a preference dataset to optimize Equation (2).

## 5 EXPERIMENTS

We report all runtimes on NVIDIA A100 GPUs accelerated using VLLM (Kwon et al., 2023) for inference and DeepSpeed (Microsoft, 2021) for training. Our implementation uses PyTorch and the Transformer Reinforcement Learning(TRL) library (von Werra et al., 2020), and we use the open-source repository by Piet et al. (2023) that implements the four surveyed watermarking methods. We test robustness using the hyper-parameters suggested by Piet et al. (2023). All LLMs used in our evaluations have been instruction-tuned. Table 1 summarizes all other surveyed evasion attacks.

### 5.1 PREFERENCE DATASET COLLECTION

For a given watermarked sequence generated by the surrogate model, the attacker generates $N$ paraphrased versions using the non-optimized paraphraser and calculates the best-of-N evasion rate with the surrogate key (Algorithm 1, lines 9-12). Figure 2 shows the number of repetitions $c$ needed to achieve a given evasion rate across four watermarking methods using `Llama2-7b` as both the surrogate and paraphrasing model. Our attacker can choose the best-of-N paraphrases because they know the surrogate watermarking key to detect a watermark. The attacker cannot choose the best-of-N paraphrases against the provider's watermarked text, as they lack access to the provider's key.

Figure 2 shows the success rate of observing at least one positive sample after N paraphrases

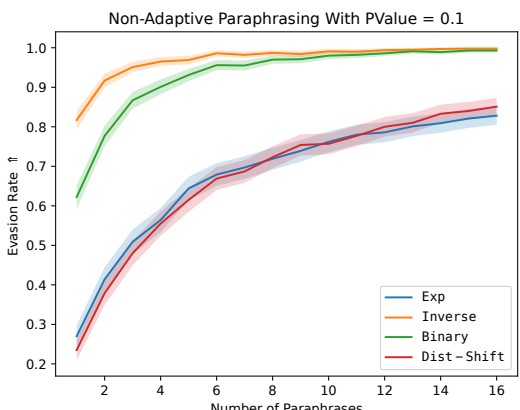

Figure 2: The expected evasion versus repetition rate with `Llama2-7b` (lines 9-12, Algorithm 1).

against methods designed for robustness (`Dist-Shift`, `Exp`) and undetectability (`Inverse`, `Binary`). The attacker requires only limited computational resources to curate a large preference dataset against any of the four surveyed watermarks. For instance, to collect $|D| = 7\,000$ preference samples of $T = 512$ tokens each, at $1\,800$ tokens/second, we expect to generate $|D|$ positive examples in approximately 1.5 GPU hours for `Dist-Shift`, but only 0.5 GPU hours for `Inverse`. In practice, when including the overhead of evaluating quality and detecting watermarks, we require less than 5 GPU hours to curate $7\,000$ samples for `Dist-Shift`. At current AWS[2] rates, an attacker faces negligible costs of less than USD 10\$ to curate a preference dataset containing $7\,000$ samples.

## 5.2 ABLATION STUDIES

In our experiments, we ablate over the following settings.

- **Adaptivity**: (*Adaptive*) The same watermarking method is used for training and testing. (*Non-adaptive*) The attack is tested against unseen watermarking methods.
- **Target Models**: We evaluate 2 models used by the provider: `Llama2-13b`, `Llama3-70b`
- **Attacker's Models**: Our attacker matches surrogate and paraphrasing models. We consider `Llama2` (Touvron et al., 2023) and `Qwen2.5` (Qwen, 2024) from 0.5b to 7b parameters.
- **Watermarking Methods**: `Exp` (Aaronson & Kirchner, 2023), `Dist-Shift` (Kirchenbauer et al., 2023b), `Inverse` (Kuditipudi et al., 2024), `Binary` (Christ et al., 2023)
- **Hyper-Parameters**: We ablate over multiple hyper-parameters that a provider can choose.
- **False Positive Rates (FPR)**: Appendix A.8 ablates over $\rho \in \{0.01, 0.025, 0.05, 0.075, 0.1\}$ when the provider can tolerate higher FPR thresholds for detection.

A watermark has been *retained* if the null hypothesis that the watermark is not present in the content can be rejected with a given p-value specified by the provider. The *evasion rate* is calculated as the fraction of watermarked text that does not retain the watermark after applying the paraphrasing attack. Due to the lack of a gold-standard metric to assess text quality, we measure quality with multiple metrics: LLM-Judge, LLM-CoT and LLM-Compare from (Piet et al., 2023), Mauve (Pillutla et al., 2021) and Perplexity (PPL) with `Llama3-8B-Instruct`. To enhance clarity, we only report the LLM-Judge metric in the main paper following Piet et al. (2023) and report other quality metrics in the Appendix. Unless otherwise specified, we use a p-value threshold of $\rho = 0.01$.

## 5.3 EXPERIMENTAL RESULTS

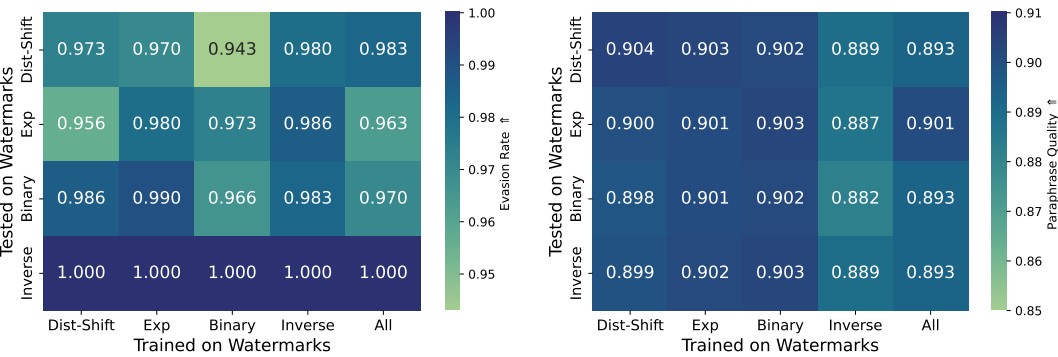

Figure 3: The evasion rates (Left) and text quality measured with LLM-Judge (Right). The attacker uses a matching `Llama2-7b` surrogate and paraphraser model versus the provider's `Llama2-13b`. We evaluate both the adaptive (diagonals) and the non-adaptive case (values not on the diagonal). For example, we obtain the bottom left value by training against `Dist-Shift` and testing on `Inverse`.

**Adaptivity**. Figure 3 shows the evasion rate and text quality of our methods trained in the adaptive and non-adaptive settings when the provider uses `Llama2-13b` and the attacker uses `Llama2-7b`.

[2]https://aws.amazon.com/ec2/capacityblocks/pricing/

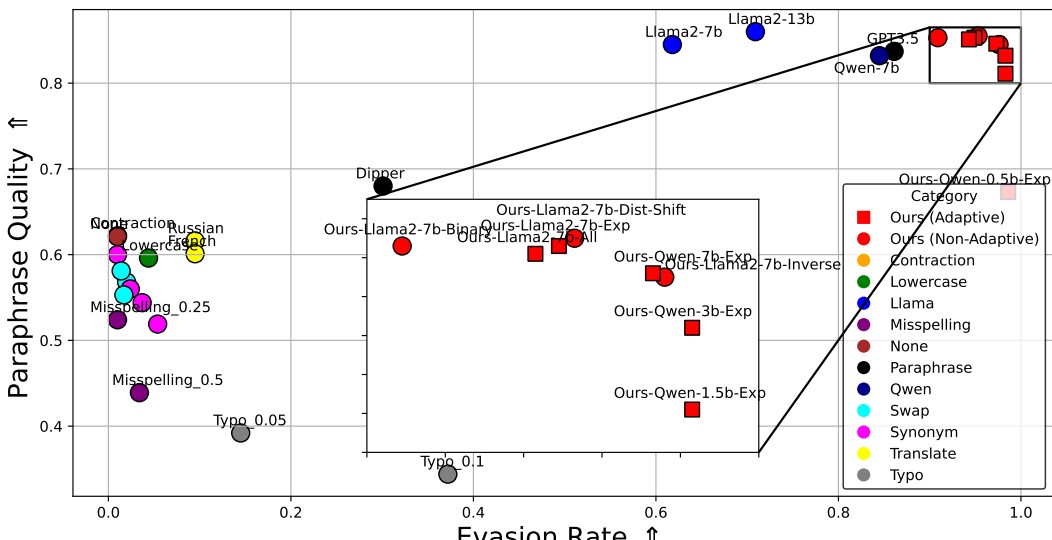

Figure 4: The evasion rate versus text quality trade-off of all surveyed attacks when the provider uses a `Llama3-70b` model and the `Exp` (Aaronson & Kirchner, 2023) watermark. The attacker uses matching surrogate and paraphrase models with parameters ranging between $0.5b$ to $7b$ from the `Qwen2.5` and `Llama2` model families. A circle and square denote non-adaptive and adaptive attacks, respectively, and our attacks are highlighted in red. For example, `Ours-Qwen-3b-Exp` means that we evaluate a `Qwen2.5` model with $3b$ parameters optimized against the `Exp` watermark.

We find that all adaptive attacks have an evasion rate of at least $96.6\%$, while the non-adaptive attacks have an evasion rate of at least $94.3\%$. We achieve the highest overall evasion rate when training against the `Exp` watermark, which achieves an evasion rate of at least $97.0\%$. We train one attack, denoted `All`, against all four surveyed watermarking methods and test it against each watermark separately. Interestingly, `All` performs slightly worse compared to training only on `Exp` as it has an evasion rate of at least $96.3\%$ and a lower text paraphrased quality of at least $0.893$ against $0.901$ when training only on `Exp`. In summary, Figure 3 shows that adaptive attacks trained against one watermark remain highly effective when tested against unseen watermarks in the non-adaptive setting.

**Model Sizes.** Figure 4 shows the Pareto front against the `Exp` watermark with a `Llama3-70b` target model. Our attacker uses paraphraser models with at most $7b$ parameters, which is less than the $11b$ DIPPER model (Krishna et al., 2023) used to test robustness currently. We make three observations:

1. **Non-adaptive baseline** attacks such as Contraction, Swapping and Synonym replacements are ineffective and have a low evasion rate of less than $20\%$.

2. **Non-adaptive model-based** paraphrasing attacks such as using vanilla `Llama2-7b` or `ChatGPT3.5` models have a substantially higher evasion rate of $61.8\%$ up to $86.1\%$ respectively. Tuning `Llama2-7b` using our approach in the non-adaptive setting improves the evasion rate substantially to $90.9\%$ (when trained on `Binary`) and up to $97.6\%$ (when trained on `Inverse`). These non-adaptive, optimized attacks have a paraphrased text quality of $0.853$ and $0.845$, slightly improving over `ChatGPT3.5`, rated only $0.837$.

3. **In the adaptive setting**, our fine-tuned `Qwen2.5-7b` achieves an evasion rate of $97.3\%$ at a slightly higher text quality of $0.846$ compared to `Llama2-7b-Inverse`.

By ablating over `Qwen2.5` between $0.5b$ and $7b$ parameters, we find that attackers can strictly improve paraphrased text quality at similar evasion rates by using more capable paraphrases with more parameters. Figure 11 in the Appendix shows results against a `Llama2-13b` target model, which are consistent with those against `Llama3-70b`. Against smaller target models, attackers can achieve higher evasion rates and text quality ratings.

**Text Quality**. Table 3 shows (i) a watermarked text sample generated using `Llama2-13b` with `Dist-Shift`, (ii) paraphrased text using a non-optimized `Llama2-7b` model, and (iii) paraphrased

text obtained with an adaptively tuned `Llama2-7b` model using our attack. We observe that all paraphrased texts preserve quality, but our attack has the lowest green-to-red token ratio (i.e., maximizes the evasion rate). Table 4 in the Appendix shows a quantitative analysis of the median quality of generated text for a vanilla `Llama2-7b` model compared to our best adaptive and non-adaptive attacks. It shows that text quality is preserved across five text quality metrics when using our attacks. We only show one paragraph of generated text that we truncated due to space restrictions and Tables 5 and 6 in the Appendix show non-truncated samples. Table 6 shows a rare case when our attack fails at evading watermark detection after paraphrasing.

**Hyper-Parameters.** We evaluate how the strength of the bias parameter used for `Dist-Shift` affects its robustness against our attacks. Our attacker does not know which hyperparameters are used by the provider. We set the bias $\beta \in \{1, 2, 4, 8\}$, where higher bias should lead to higher robustness (Piet et al., 2023; Kirchenbauer et al., 2023b). We train our attacks once with the $\beta = 4$ value suggested by Piet et al. (2023) and test it against all other hyper-parameters. Table 2 shows that our adaptive and non-adaptive attacks remain the most effective across all hyper-parameters.

| $\beta$ | Dist-Shift | Llama2-7b | Llama2-7b-Exp | Llama2-7b-Dist-Shift |
|---|---|---|---|---|
| 1 | 0.94, 0.72 | 0.94, 0.96 | 0.94, 0.98 | 0.95, **0.99** |
| 2 | 0.94, 0.20 | 0.95, 0.90 | 0.95, **0.98** | 0.95, **0.98** |
| 4 | 0.95, 0.00 | 0.96, 0.67 | 0.94, **0.97** | 0.94, **0.97** |
| 8 | 0.71, 0.00 | 0.92, 0.60 | 0.94, 0.95 | 0.94, **0.96** |

Table 2: We study our attack's effectiveness for different bias hyper-parameters for the `Dist-Shift` (Kirchenbauer et al., 2023a) watermark. The top row label refers to the generation method, and we show two values: The text quality and the evasion rate at 512 tokens.

**Adaptive vs Non-adaptive.** Figure 5 shows two results to compare the non-optimized `Llama2-7b` with our adaptively tuned `Llama2-7b` model. The result on the left plots the cumulative density of p-values. Our method strictly improves over the non-optimized model as it generates paraphrased text with higher mean p-values for watermark detection. The result on the right plots the expected p-value against the token length. The watermarked text has a median p-value of less than $0.01$ after approximately 170 tokens, whereas the non-optimized `Llama2-7b` model has an expected p-value of $0.10$ at around 500 tokens compared to an expected p-value of 0.4 for our adaptively tuned model.

**Additional Testing.** We present more results to compare adaptive versus non-adaptive attacks in Appendix A.3, including tests against other recently released watermarking methods. These results are consistent with our findings in the main part of the paper that adaptive attacks are Pareto optimal and outperform much larger, closed-source systems such as GPT4o at evading watermark detection. We kindly refer the reader to Appendix A.3 for more baseline tests and Appendix A.4 for an analysis of the impact of paraphrasing on the top-50 token distribution.

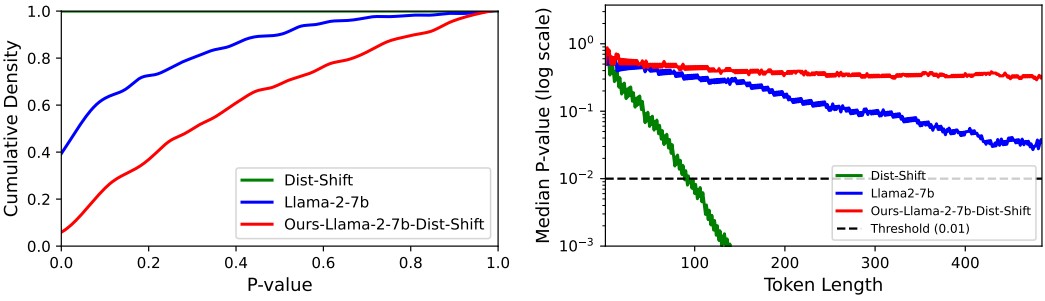

Figure 5: (Left) The cumulative density of p-values on the `Dist-Shift` watermark (blue), a vanilla `Llama2-7b` paraphraser (green) and our adaptively tuned `Llama2-7b` paraphraser (red). (Right) The median p-value relative to the text token length with a threshold of $\rho = 0.01$ (dashed line).

| `Llama2-13b-Dist-Shift` (163 green and 36 red) |
| --- |
| Cervantes' "Don Quixote" is a groundbreaking masterpiece of literature that has enthralled readers for over 400 years. The novel tells the story of Alonso Quixano, an old man who becomes so consumed by the chivalrous tales of his day that he renames himself Don Quixote and embarks upon an adventurous journey filled with humor, satire, and timeless themes about the human condition.↩↩ The story follows Quixote and his trustworthy sidekick Sancho Panza, a simple and pragmatic squire who provides moments of humor and contrast to the lofty ideals of the delusional Quixano/Don Quixote. Their travels in rural Spain lead them to encounter all sorts of people they mistake for knights, ladies in distress, bandits, and giants who are in fact ordinary villagers going about their everyday lives. |

| `Llama2-7b` (125 green and 69 red) |
| --- |
| "Don Quixote" is a groundbreaking masterpiece of literature that has captivated readers for over 400 years. The novel tells the story of Alonso Quixano, an elderly man whose fixation on chivalrous tales leads him to change his name to Don Quixote and embark on a thrilling adventure replete with humor, satire, and timeless themes concerning the human condition.↩↩ The narrative follows Don Quixote and his loyal squire Sancho Panse, a practical and good-natured individual who provides moments of levity and contrast to the lofty ideals of the delusional Quixano/Don Quixote. Their travels across rural Spain result in encounters with various people who they misidentify as knights, distressed ladies, bandits, and ordinary villagers going about their daily lives. |

| `Ours-Llama2-7b-Dist-Shift` (81 green and 78 red) |
| --- |
| Spain has been blessed with a literary treasure that has captivated readers for over four centuries in the form of Miguel de Cervantes' immortal masterpiece, "Don Quixote". This timeless tome weaves a hilarious and poignant tale of a man consumed by the chivalrous stories of his youth, who renames himself Don Quixote and sets forth on a journey filled with moments of satire and a piercing examination of the human condition.↩↩As Don Quixote and his trusty squire Sancho Panza traverse rural Spain, they encounter various unsuspecting villagers, mistaking them for knights, maidens in distress, bandits, and even giants. |

Table 3: (Top) Watermarked text from the provider's `Llama2-13b` model, (Middle) a paraphrased version from a vanilla `Llama2-7b` model, and (Bottom) paraphrased text using our adaptively tuned `Llama2-7b` model. Green/red indicates whether a token is watermarked. A lower green-to-red token ratio implies a higher evasion rate. Due to space constraints, we only show truncated texts. Tables 5 and 6 in the Appendix show entire samples with up to 512 tokens.

## 6 DISCUSSION

**Effectiveness of Adaptive Attacks.** Our work demonstrates that content watermarking methods for large language models are vulnerable to adaptively optimized attacks. Attackers can adaptively fine-tune relatively small open-weight models, such as `Llama2-7b` (Touvron et al., 2023), using less than seven GPU hours to evade watermark detection from substantially larger and more capable open-source models, such as `Llama3.1-70b` (Dubey et al., 2024). Our attacks remain effective even in the non-adaptive setting when testing with unseen watermarking methods. Our findings challenge the robustness claims of existing watermarking methods, and we propose improved methods to test robustness using adaptive attacks.

**Analysis.** Studying *why* adaptive attacks work is challenging due to the non-interpretability of the optimization process. The ability to maximize Equation (2) implies the ability to evade detection since Equation (2) encodes robustness for any watermarking method. The effectiveness of non-adaptive attacks could be explained by the fact that all surveyed watermarks are similar in that they operate on the token level. Hence, an effective attack against one watermark likely generalizes to other unseen watermarks. Adaptive attacks further improve effectiveness as there are at least three learnable signals for paraphrasing watermarked text: (1) Avoid repeating token sequences as they likely contain the watermark, (2) find text replacements with low impact in text quality to maximize evasion rate (e.g., uncommon words or sentence structures), and (3) calibrate the minimum token edit distance and lexical diversity that on average (over the randomness of the key generation process) evades detection. We refer to Appendix A.5 for a more detailed analysis of our approach's effectiveness.

**Attack Runtime.** Our attacks involve two steps: Dataset Curation and Model Optimization. Curating 7 000 samples requires less than 5 GPU hours, and model optimization requires only approximately 2 GPU hours for a `Llama-7b` model at 16-bit precision. These attacks can be executed with limited computational resources and cost less than USD 10$ with current on-demand GPU pricing.

**Online Attacks.** Our work focuses on *offline* attacks that do not require access to the provider's watermark detection functionality. Offline attacks help study the robustness of a watermarking method without any information about the specific secret key generated by the provider. An *online* attacker can learn information about the provider's secret key through accessing `Verify`, which reduces the attack's uncertainty and could substantially improve the attack's effectiveness further.

**Limitations.** Our study also did not focus on evaluating adaptive defences that could be designed against our adaptive attacks. Adaptive defences have not yet been explored, and we advocate studying their effectiveness. We believe our optimizable, adaptive attacks will enhance the robustness of future watermarking methods by including them in their design process, for instance, by using adversarial training. We focused exclusively on text generation tasks and did not explore other domains, such as source code generation or question-answering systems where different text quality metrics may be used to evaluate an attack's success. We did not consider the interplay between watermarking and other defences, such as alignment or content filtering, which could collectively control misuse.

We acknowledge that LLM-as-a-Judge is an imperfect and noisy metric that may not align with human judgment. In the main part of our paper, we use `Llama3-8B`-as-a-Judge since this metric is easily reproducible. Appendix A.3 shows results using GPT4-mini-as-a-Judge that are consistent with our findings. More work is needed to study the metric's alignment with human judgment.

## 7 RELATED WORK

We evaluate the robustness of *content* watermarking (Lukas & Kerschbaum, 2023) methods against no-box, offline attackers in the adaptive and non-adaptive settings (see Section 2). Other watermark evasion attacks (Hu et al., 2024; Kassis & Hengartner, 2024; Lukas et al., 2024) focus on the image domain, whereas we focus on LLMs. Jovanović et al. (2024); Pang et al. (2024) propose black-box attacks against LLMs that require collecting many watermarked samples under the same key-message pair. We focus on no-box attacks. Jiang et al. (2023) propose online attacks with access to the provider's watermark verification, whereas we focus on a less capable *offline* attacker who cannot verify the presence of the provider's watermark. Current attacks are either non-adaptive, such as DIPPER (Krishna et al., 2023) or handcrafted against one watermark (Nicks et al., 2024). We focus on optimizable, adaptive attacks and show that they remain effective in the non-adaptive setting.

## 8 CONCLUSION

Our work demonstrates the vulnerability of current LLM watermarking methods to adaptive attacks. We propose preference-based optimization on open-weight models to undermine the robustness of four watermarking methods and find that relatively small models such as `Llama2-7b` can be adaptively tuned to evade detection from larger models, such as `Llama3-70b`. Our adaptively tuned attacks outperform all existing attacks and can be trained using negligible computational resources of less than seven GPU hours. Adaptively tuned attacks remain effective in the non-adaptive setting against unseen watermarks. Our findings challenge the security claims of existing watermarking methods and suggest that future defences must consider adaptive attackers when testing robustness.

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

## A  APPENDIX

### A.1  QUALITY METRICS

Ideally, to evaluate the quality of an LLM-generated text, one would need a set of human evaluators, each giving their own score according to a certain rubric, and then have all the scores aggregated. However, this is impractical to achieve, both for the attacker and the defender. Therefore, we employ multiple surrogate metrics from the literature: LLM-Judge, LLM-CoT and LLM-Compare from (Piet et al., 2023), Mauve (Pillutla et al., 2021) and Perplexity (PPL) with `Llama3-8B-Instruct`. Note that all of these are implemented in the `MarkMyWords (MMW)` (Piet et al., 2023) benchmark utilized for our experiments. All the metrics evaluate a response (whether watermarked, or pertrubed sample) against a baseline (either the original prompt, a non-watermarked sample or the model's logit-distribution). Below is a description of each metric, along with an indication of whether higher or lower values are better for that metric.

**LLM-Judge:** LLM-Judge directly directly prompts an instruction-tuned large language mode to evaluate the quality of a certain response with respect to its original prompt. The response is decoded greedily (temperature =0) to ensure deterministic results. Criteria evaluated include:accuracy, level of details, and typographical, grammatical and lexical correctness. A higher score is better. For this, we use `Llama3-8B-Instruct` to evaluate and the following prompt:

> **LLM-Judge Prompt**
>
> ```
> You are given a prompt and a response, and you provide a grade
> out of 100 measuring the quality of the response, in terms of
> accuracy, level of details, and typographical, grammatical and
> lexical correctness.  Remove points as soon as one of the criteria
> is missed.  Prompt:{}, Response:{}
> ```

**LLM-CoT (Chain-of-Thought):** LLM-CoT evaluates the quality of the watermarked/attacked responses using CoT-based reasoning Wei et al. (2023). A higher score is better. For this, we also use `Llama3-8B-Instruct` to evaluate and the following prompt:

> **LLM-CoT Prompt**
>
> ```
> Please act as an impartial judge and evaluate the quality of
> the response provided by an AI assistant to the user question
> displayed below.  Your evaluation hinges on the following factors:
> accuracy, level of details, typographical correctness, grammatical
> correctness and lexical correctness.  Be as objective as possible,
> and account for every one of these criteria in our evaluation.
> Begin your evaluation by providing a short explanation.  After
> providing your explanation, please rate the response on a scale
> of 1 to 100 by strictly following this format:  "[[rating]]", for
> example:  "Rating:  [[51]]".  Your rating must come AFTER your
> explanation, not before.  Prompt:{}, Response:{}
> ```

**LLM-Comparator:** The LLM-Comparator is used to compare non-watermarked baseline response and the watermarked or attacked response. 0 indicates that the non-watermarked response is better, 0.5 indicates a tie, and 1 shows that the watermarked or attacked response is better. For this, we also use `Llama3-8B-Instruct` to evaluate and the following prompt:

**MAUVE.** MAUVE measures the similarity between two text distributions. In our case, the two distributions are the non-watermarked baseline response and the watermarked/paraphrased response. Higher MAUVE scores indicate that both texts match their content, quality and diversity. MAUVE is computed with the Kullback–Leibler (KL) divergences between two distributions in a lower-dimensional latent space. It correlated with human evaluations over baseline metrics for open-ended text generation Pillutla et al. (2021). We use the `gpt2-large` model to compute the MAUVE score in our implementation.

**Perplexity (PPL):** Perplexity is a common language modelling metric that quantifies how well a model predicts a text sample. It is calculated based on the probability that the model assigns to a sequence of words. Lower perplexity values indicate that the model is more confident and accurate in its predictions, making lower scores better for this metric.

Table 4 shows the median text quality metrics to compare the vanilla `Llama2-7b` paraphraser to our best adaptive and non-adaptive attacks against the `Llama2-13B` and `Llama3-70B` target models. The table shows that our attacks have similar quality to the vanilla `Llama2-7b` paraphraser across the board. Our attacks have a higher MAUVE score, indicating that our text is closer to the non-watermarked text than the vanilla `Llama2-7b` paraphraser. The higher perplexity is not a concern, as it just shows that the large language model does not expect the text.

| Target: `Llama2-13B` | LLM-Judge ⇑ | LLM-CoT⇑ | LLM-Compare⇑ | Mauve⇑ | PPL⇓ |
|---|---|---|---|---|---|
| `Llama2-7b` | 0.92 | 0.85 | 0.00 | 0.17 | 4.74 |
| Ours-Best-Adaptive | 0.92 | 0.85 | 1.00 | 0.42 | 6.69 |
| Ours-Best-Non-Adaptive | 0.92 | 0.85 | 0.50 | 0.37 | 6.32 |
| Target: `Llama3-70B` | | | | | |
| `Llama2-7b` | 0.95 | 0.72 | 0.00 | 0.22 | 4.84 |
| Ours-Best-Adaptive | 0.95 | 0.72 | 0.50 | 0.55 | 6.10 |
| Ours-Best-Non-Adaptive | 0.95 | 0.72 | 0.50 | 0.31 | 6.15 |

Table 4: Various median text quality metrics to compare the vanilla `Llama2-7b` paraphraser to our best adaptive and non-adaptive attacks. We limit all attacks to at most 7b parameter models.

## A.2 DATASET CURATION

We generate a synthetic prompt dataset spanning various topics, including reviews, historical summaries, biographies, environment, science, math, fake news, recipes, travel, social media, arts, social

sciences, music, engineering, coding, sports, politics, health, and more. The dataset has 1000 prompts and is collected by repeatedly prompting a large language model (ChatGPT-4) to generate topic titles. We then wrap combinations of these titles in prompts. We intend to release the dataset for the public, but it should be very easy to replicate.

For every prompt, we generate watermarked output from all watermarks; then, we use that as input to our paraphrasers. Each paraphraser is to generate 16 paraphrases for each input. We then filter these paraphrases as per Algorithm 1 to create the training preference pairs. Larger models have higher quality output and so have a higher yield of successful paraphrases. We use the same number of paraphrases for each model, even when it generates different yields.

Figure 6 shows the expected evasion rate versus the number of paraphrases ablated over varying model sizes of Qwen2.5 (Qwen, 2024) against the Exp watermark. We find that the expected evasion rate increases with the number of paraphrases, but the rate of increase diminishes as the number of paraphrases increases. We find that the expected evasion rate does not improve significantly close to 16 paraphrases and that bigger models tend to have higher evasion rates for the same number of paraphrases. An exception to this is the 1.5b model which surprisingly performs very well (better than the 3b) for the same number of paraphrases. This, however, could be due to different pretraining parameters of the base model or other factors.

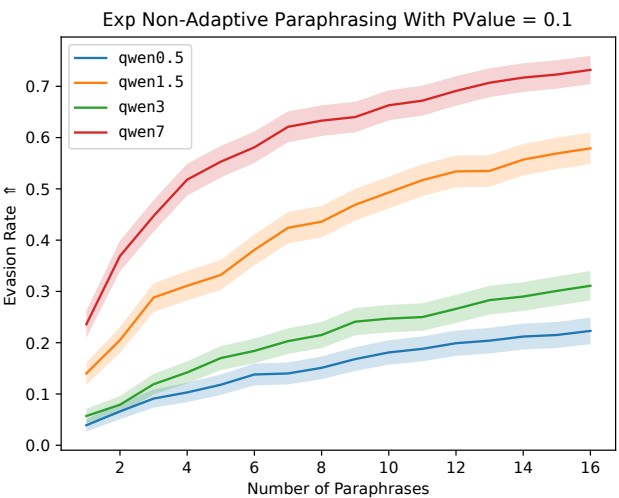

Figure 6: The expected evasion rate versus the repetition rate ablated over varying model sizes of Qwen2.5 (Qwen, 2024) against the Exp watermark (Algorithm 1, lines 9-12). Shaded areas denote 95% confidence intervals.

## A.3 BASELINE TESTING AGAINST OTHER WATERMARKS

We include more robustness tests against recently released watermarks such as SynthID (Dathathri et al., 2024), Unigram (Zhao et al., 2024) and SIR (Liu et al., 2024). We refer to the author's papers for detailed descriptions of these watermarks. GPT4o is part of the Pareto front against only SIR and KGW due to its high text quality and low evasion rates of less than 90%. It is not part of the Pareto front against SynthID, EXP and Unigram, where only our attacks are part of the Pareto front. While it may be possible to use better prompts for GPT4o to achieve a higher text quality or evasion rate, there are other limitations when using closed systems to evade detection.

1. Their usage can be expensive as the user is typically charged per token.

2. The system could embed its own watermark into paraphrased text.

3. There could be guardrails such as safety alignments which prevent these systems from arbitrarily paraphrasing text.

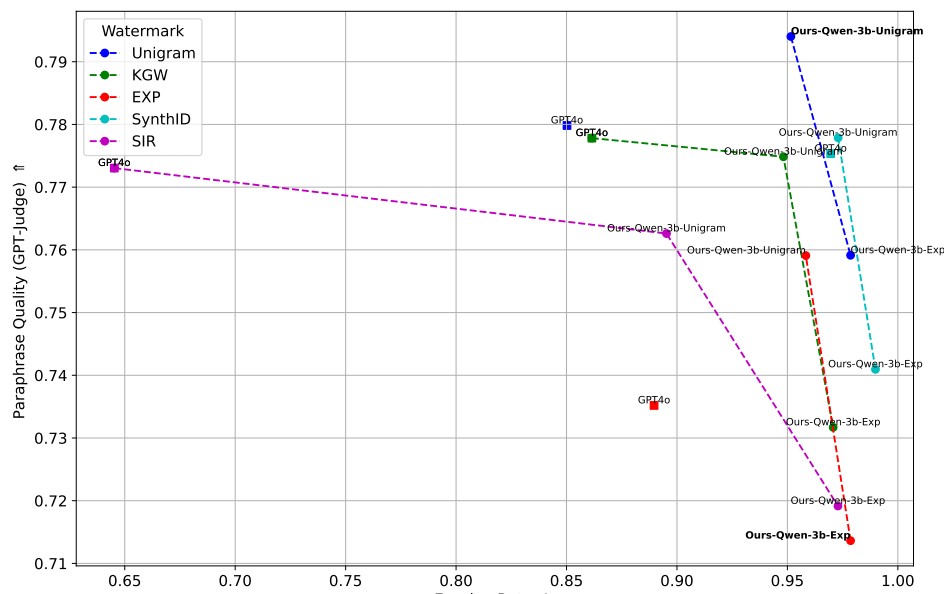

Figure 7: Additional results using Qwen-3b against KGW and EXP, which we study in the main part of the paper, and more recently released watermarks such as SynthID (Dathathri et al., 2024), Unigram (Zhao et al., 2024) and SIR (Liu et al., 2024). Dashed lines denote the Pareto front, and we highlighted adaptively trained attacks in bold. We used GPT4o's version from November 23rd, 2024. The y-axis uses GPT4-mini as a judge, and the x-axis shows the evasion rate.

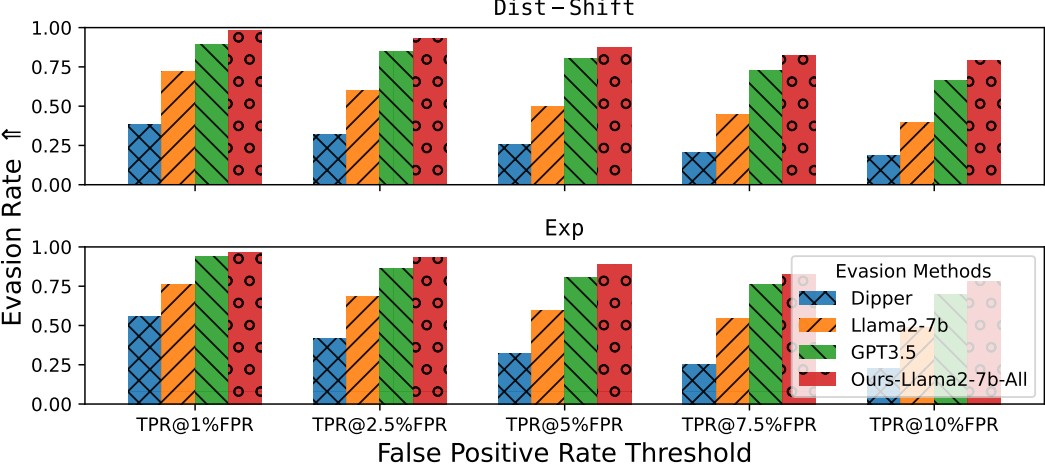

Figure 8: The evasion rates against a watermarked `Llama2-13b` model. We compare non-adaptive attacks, including ChatGPT3.5, versus our adaptively fine-tuned `Llama2-7b` paraphraser model.

In contrast, our method allows working with relatively small open-weight models that adversaries can fully control.

### A.4    TOKEN DISTRIBUTION

**Text Quality.** Appendix A.4 shows the top-50 token distribution that appear in the watermarked text. We compare it with the token frequency in the paraphrased text using as paraphrasers (i) GPT4o, (ii) a baseline Qwen-3b model and (iii) our adaptively tuned Qwen3b model against the Unigram watermark Zhao et al. (2024). We observe that all paraphrasers have a similar token distribution and

that across all three paraphrasers, on average, the top 50 tokens appear less frequently than in the original, watermarked text. The largest difference we observe between the baseline Qwen3b and our adaptively tuned model are the frequencies of the tokens 'The' and ' ' (space between words), which our model uses less frequently. Compared to GPT4o, the baseline Qwen-3b model uses some tokens, such as ' As', less frequently, while other tokens, such as ' but', appear more frequently.

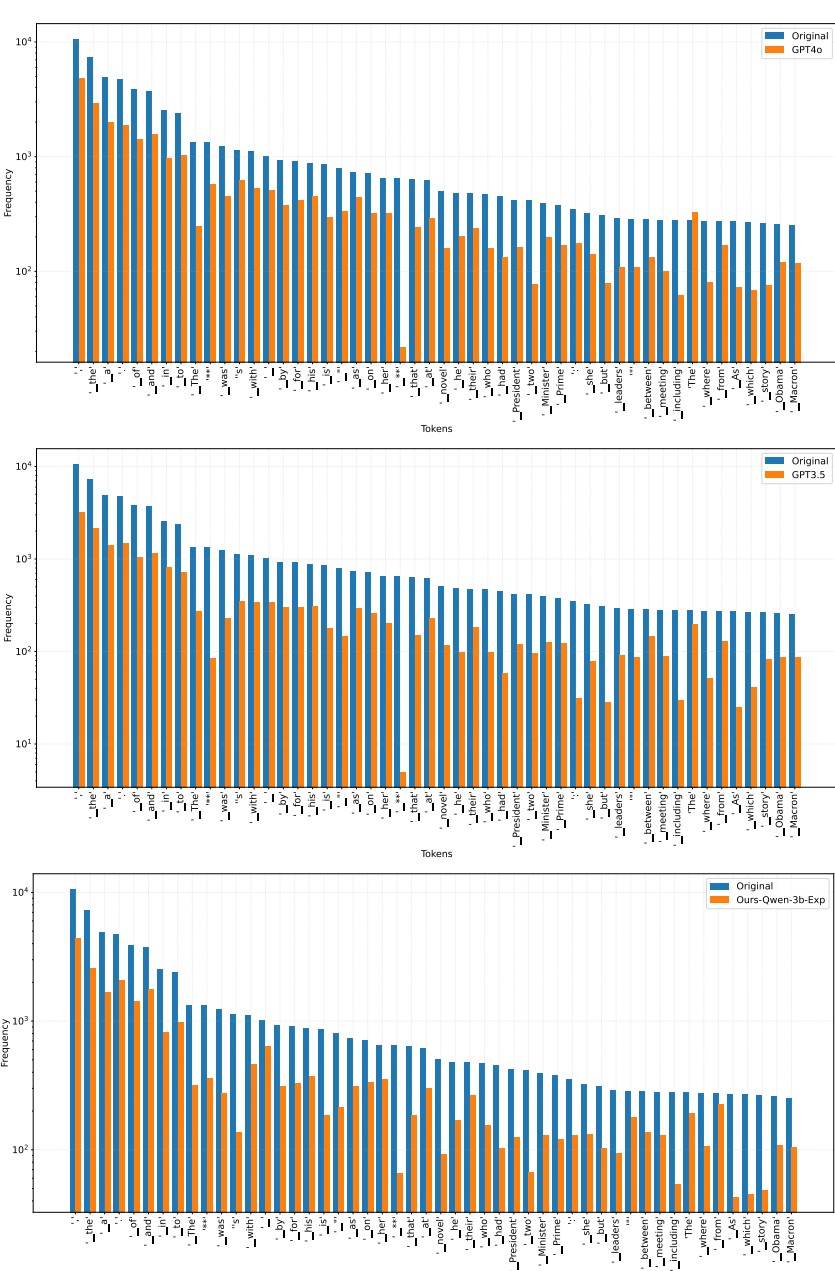

Figure 9: An analysis of the top-50 tokens in paraphrased text generated with the Unigram watermark (Zhao et al., 2024), using as a paraphraser (top) GPT4o, (center) an off-the-shelf Qwen-3b model and (bottom) our adaptively tuned Qwen-3b model.

## A.5 DETAILED TEXTUAL ANALYSIS

Our goal is to further analyze why our adaptively tuned paraphraser better evades detection than other approaches. We begin by studying the overlap of N-grams between the watermarked and paraphrased

texts, which we call the N-gram overlap ratio between two sequences $x_1, x_2 \in \mathcal{V}^*$.

$$N_g(x_1, x_2, n) = \frac{|\text{ngrams}(x_1, n) \cap \text{ngrams}(x_2, n)|}{|\text{ngrams}(x_1, n) \cup \text{ngrams}(x_2, n)|} \quad (3)$$

The 'ngrams' function tokenizes a sequence and returns the set of n-grams. The N-gram overlap ratio is always between [0,1]. A high overlap for a given $n \in \mathbb{N}$ states that the same N-grams appear in both sequences. Since the surveyed watermarks operate on a token level, a low overlap ratio would suggest a high evasion rate. We also evaluate the token edit distance ratio between two sequences, which is calculated as follows:

$$L(x_1, x_2) = \frac{\text{Levenshtein}(x_1, x_2)}{\text{len}(x_1) + \text{len}(x_2)} \quad (4)$$

The token edit distance calculates the Levensthein distance between two sequences. Note that the N-gram overlap ratio is calculated over sets of N-grams. In contrast, the Levenshtein distance is calculated over (ordered) sequences, meaning that the position of the token matters. A high Token Edit Distance ratio suggests that two texts do not have the same tokens at the same positions in the sequence, which also suggests a higher evasion rate.

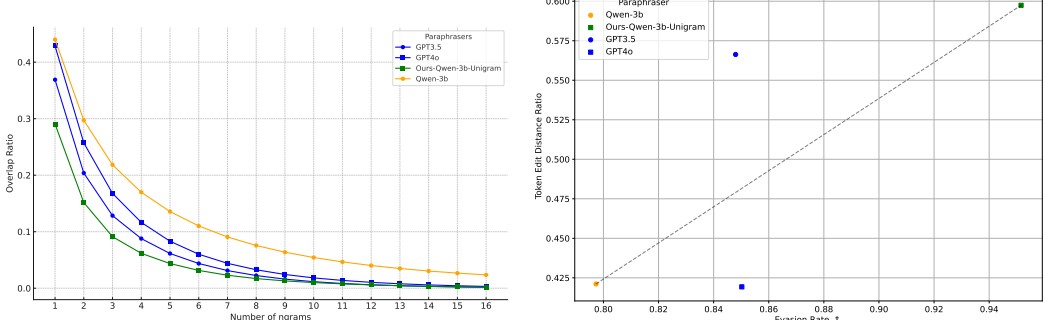

Figure 10: (Left) The N-gram overlap ratio between watermarked text and text paraphrased by (i) GPT3.5, (ii) GPT4o, (iii) our adaptively tuned Qwen-3b paraphraser and (iv) a baseline Qwen-3b paraphraser. The overlap is calculated as the number of N-grams in the paraphrased text that also appear in the watermarked text divided by the total number of N-grams in the watermarked text. Lower overlap means that both texts are *less* similar. (Right) We plot the evasion rate against the normalized token edit distance between paraphrased and watermarked text using different paraphrasers. The dashed line represents the difference between the non-optimized Qwen-3b paraphraser and our adaptively tuned Qwen-3b paraphraser.

**Results.** Figure 10 (left) shows the N-gram overlap ratio between watermarked text and the text produced by four paraphrasing methods. We observe that across all N-grams, our adaptive paraphraser achieves the lowest overlap ratio. Figure 10 (right) shows the mean token edit distance ratio between watermarked and paraphrased text in relation to the evasion rate. We observe that the non-optimized, baseline Qwen-3b model has a low token edit distance ratio and a low evasion rate. In contrast, our adaptively tuned model has a much higher evasion rate and a high token edit distance ratio. These findings suggest that our adaptive optimization process learned to increase the mean token edit distance and minimize the overlap ratio to maximize evasion rates while preserving text quality.

## A.6 WATERMARK PARAMETERS

To select the optimal parameters for the watermarking methods, we follow the guidelines provided by Piet et al. (2023). We use a key length of 4 for all watermarks and a text-dependent sliding window randomness of size 3. We set the skip-probability to 0.025 for all watermarks except for the Dist-Shift watermark, where we set it to 0. Skip-probability is a technique that randomly skips the watermarking selection procedure for some tokens to allow more diverse generation and it works best with schemes that can be made indistinguishable, like the Exp, Binary and Inverse

watermarks. We also use the optimal temperature for every watermark (1.0 for all except for the Dist-Shift watermark, where we use 0.7). Specific to the Dist-Shift watermark, we use the suggested green-red list ratio $\gamma$ of 0.5 and a bias parameter $\beta$ of 4.

## A.7    ATTACK DESCRIPTION

**Prompting.** We use the following prompt to train our paraphraser models. The prompt is adapted from Kirchenbauer et al. (2023b). Additionally, we prefill the paraphrase answer with the text `[[START OF PARAPHRASE]]` to ensure that the model starts generating the paraphrase from the beginning of the response. During dataset curation, training and testing, we set the temperature to 1.0 to diversify the generated paraphrases.

---

**Paraphrase Prompt**

```
You are an expert copy-editor.  Please rewrite the following text in
your own voice and paraphrase all sentences.  Ensure that the final
output contains the same information as the original text and has
roughly the same length.  Do not leave out any important details
when rewriting in your own voice.  Do not include any information
that is not present in the original text.  Do not respond with a
greeting or any other extraneous information.  Skip the preamble.
Just rewrite the text directly.
```

---

**Training Hyperparameters** We train our paraphraser models using the following hyperparameters: batch size of 32, learning rate of $5 \times 10^{-4}$, and a maximum sequence length of 512 tokens. We use the AdamW optimizer with a linear learning rate scheduler that warms up the learning rate for the first 20% of the training steps and then linearly decays it to zero. We train the models for 1 epoch only to prevent overfitting. We utilize Low-Rank Adaptation (LoRA) (Hu et al., 2022) to reduce the number of trainable parameters in the model. We set the rank to 32 and the alpha parameter to 16.

## A.8    ADDITIONAL ABLATION STUDIES

**False Positive Rates.**    Figure 8 shows the detection rates at different FPR-thresholds $\rho \in \{0.01, 0.025, 0.05, 0.075, 0.1\}$ against the Dist-Shift and Exp watermarking methods. We focus on these two methods as they are more robust than Inverse and Binary. Our results show that across all evaluated FPR thresholds, our adaptive attacks outperform all other surveyed attacks against both watermarking methods. If the provider tolerates a 10% FPR, our adaptive attacks achieve an evasion rate of only 80% and 77% against Dist-Shift and Exp, respectively.

## A.9    EXTRA TABLES AND FIGURES

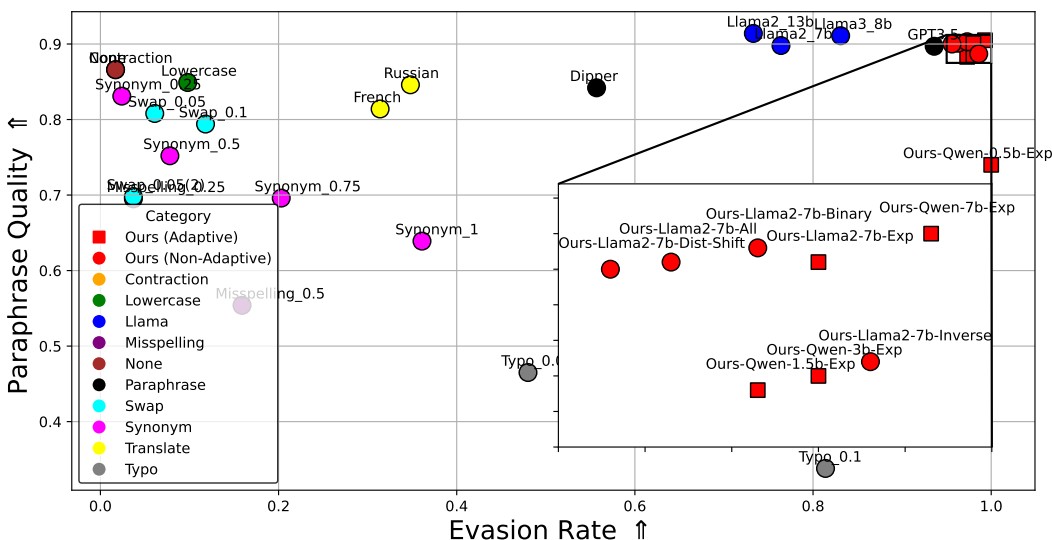

Figure 11: The evasion rate versus text quality trade-off of all surveyed attacks when the provider uses a `Llama3-13b` model and the `Exp` (Aaronson & Kirchner, 2023) watermark. The attacker uses matching surrogate and paraphrase models with parameters ranging between $0.5b$ to $7b$ from the `Qwen2.5` and `Llama2` model families. A circle and square denote non-adaptive and adaptive attacks, respectively, and our attacks are highlighted in red. For example, `Ours-Qwen-3b-Exp` means that we evaluate a `Qwen2.5` model with 3b parameters optimized against the `Exp` watermark.

`Llama2-13b-Dist-Shift` (463 green vs 41 red tokens)

Cervantes' "Don Quixote" is a groundbreaking masterpiece of literature that has enthralled readers for over 400 years. The novel tells the story of Alonso Quixano, an old man who becomes so consumed by the chivalrous tales of his day that he renames himself Don Quixote and embarks upon an adventurous journey filled with humor, satire, and timeless themes about the human condition.←→← The story follows Quixote and his trustworthy sidekick Sancho Panza, a simple and pragmatic squire who provides moments of humor and contrast to the lofty ideals of the delusional Quixano/Don Quixote. Their travels in rural Spain lead them to encounter all sorts of people they mistake for knights, ladies in distress, bandits, and giants who are in fact ordinary villagers going about their everyday lives.←→← One of the most profound and enduring elements of the novel's themes is the conflict of reality and perception. Quixote, driven by the chivalrous books he reads and his own imagination, mistakes windmills for giants, a puppet play as a real tale of love and betrayal, and a barber from a nearby village who is dressed in his Sunday best but Quixote sees as the Grand Duchess of Lithuania. Through these absurd but comedic misadventures, Cervantes creates a timeless commentary on the nature of truth, reality, and the danger of letting our imaginations run too wild. Don Quixote's journey also explores issues of class and nobility as he views his own lowly status as unknightly, while the peasants and traveling players he encounters view him with suspicion or indifference. Through these contrasts, Cervantes pokes fun at the social order and the idealized notion of chivalry.← Don Quixote has been praised for its realistic portrayal of human nature, including its weaknesses and fallibilities as well as the timeless wisdom of Cervantes' observations on society in late sixteenth-century Spain. At its core, the novel is an exploration of the human capacity to dream, delusions,

`Llama2-7b` (301 green vs 201 red tokens)

"Don Quixote" is a groundbreaking masterpiece of literature that has captivated readers for over 400 years. The novel tells the story of Alonso Quixano, an elderly man whose fixation on chivalrous tales leads him to change his name to DonQuixote and embark on a thrilling adventure replete with humor, satire, and timeless themes concerning the human condition.←→← The narrative follows Don Quixote and his loyal squire Sancho Panse, a practical and good-natured individual who provides moments of levity and contrast to the lofty ideals of the delusional Quixano/Don Quixote. Their travels across rural Spain result in encounters with various people who they misidentify as knights, distressed ladies, bandits, and ordinary villagers going about their daily lives.←→← One of the most enduring aspects of the novel's themes is the conflict between reality and perception. Quixote, driven by the chivalrous books he reads and his imagination, mistakes wind mills for giants, a puppet show for a real tale of love and betrayal, and a peasant in his Sunday best for the Grand Duchess of Lithuania. Through these absurd yet comedic misadventures, Cervantes offers a timeless commentary on the nature of truth, reality, and the dangers of allowing our imaginations to run wild. Don Quixote's journey also explores issues of class and nobility as he views his lowly status as unknightly, while the peasants and traveling players he encounters view him with suspicion or indifference. Through these contrasts, Cervantes pokes fun at the social order and the idealized notion of chivalry.←→←Don Quixote has been praised for its realistic portrayal of human nature, including its weaknesses and fallibilities, as well as the timeless wisdom of Cervantes' observations on society in late 16th-century Spain. At its core, the novel is an exploration of the human capacity to dream, delude oneself, and confront reality, ultimately revealing the limitations and struggles of the human experience.

`Ours-Llama2-7b-Dist-Shift` (201 green vs 240 red tokens)

Spain has been blessed with a literary treasure that has captivated readers for over four centuries in the form of Miguel de Cervantes' immortal masterpiece, "Don Quixote". This timeless tome weaves a hilarious and poignant tale of a man consumed by the chivalrous stories of his youth, who renames himself Don Quixote and sets forth on a journey filled with moments of satire and a piercing examination of the human condition.←→←As Don Quixote and his trusty squire Sancho Panza traverse rural Spain, they encounter various unsuspecting villagers, mistaking them for knights, maidens in distress, bandits, and even giants. Through these absurd yet poignant events, Cervantes deftly explores the blurred lines between reality and perception, highlighting the dangers of allowing our imaginations to run wild. The novel's themes of truth, class, and nobility are also expertly woven throughout the narrative, as Don Quixote's lowly status is juxtaposed with the condescending views of the peasants and traveling players he encounters.←→← Throughout the novel, Cervantes offers a biting commentary on the social hierarchy of his time, while also providing a realistic depiction of human nature, complete with its frailties and limitations. At its core, "Don Quixote" is a thought-provoking exploration of the human capacity for imagination, delusion, and the ever-elusive pursuit of truth.←→←Endlessly praised for its timeless wisdom and biting social commentary, "Don Quixote" has captivated readers for centuries, leaving a lasting impact on literature and our understanding of the human experience. Its enduring themes and characters have become ingrained in our collective consciousness, reminding us of the importance of humility, empathy, and the boundless power of the human imagination.

Table 5: Watermarked text and paraphrases from different models. Green/red indicates whether a token is watermarked. The last column shows the detection confidence over the entire text.

```
Llama2-13b-Dist-Shift (474 green vs 36 red tokens)
```

Ok , here goes! ↩ ↩ Once upon a time, there was a scholar named Dr. Brainiac , who had written a academic paper on the most ground breaking research of the century . Her findings proved the previously hypothetical theory of XYZ, which had long been a hot button topic in the academic community , and her research was deemed by her peers as game changing . ↩ ↩ However ... (you knew there was going to be a but , didn't ya? )... Dr . Brainiac hit a snag . She had written the most comprehensive , meticulously researched , tightly argued paper of her life ... but she couldn't get it published! ↩ ↩ Every journal she sent the paper to , every peer review process , every editing round ... the same result. Reject! Reject! ReJECT! (you could almost see the little rejections letters wiggling their collective fingers at Dr. Brainiac ). ↩ ↩ Dr . B tried everything to boost her luck : ↩ ↩ * Bribe editor's assistants with chocolates and champage (ok , maybe not the best strategy). ↩ * Asked her cat , Prof. Meowington, to lick the pages of the manuscript (um , that didn't go well either) ↩ ↩ Baffled by their lack of progress, Dr , B took a step back to reassess the situation. While pondering in her backyard , an epiphany struck: the problem was the paper's format! It was too traditional, to bland , too ... academic! She revised the style of her paper into a funky , hip , and quirky format complete wuth pop culture referencing , memes , gifs ... and taadaahhh! It was accepted by every journal she submitted to , all on the same day. ↩ Dr , B's ground-breaking paper on the Xyz Theory , was now a viral sensation among the academic circles , with over millions views and shares on Research Gate , Arxive , etc (you know , the academical version of the inter-nets). ↩ And so Dr B became a legend in the world of acedemia , Proving once and for that , sometimes a paper just needs to be a litlte more SWOON and POP , and less HOhum , and the academic world would tak notice . The end.

```
Llama2-7b (218 green vs 222 red tokens)
```

Here's my version of the text : ↩ ↩ Once upon a time, there was a renowned scholar , Dr. Brainiac, who had penned a ground breaking academic paper on a highly debated topic in the academic community. Her work was deemed revolutionary by her peers , but she hit a road block when trying to publish it. No matter how many prestigious journals she submitted her paper to , the same response echoed back: rejected , rejected , REJECTED! Dr. Brainiac's persistence was met with utter rejection. ↩ ↩ With her reputation on the line, Dr. Brainiac contemplated desperate measures to break the deadlock. She even enlisted the help of her feline colleague , Prof. Meowington, to lend a paw to the editing process, but alas, it seemed the paper was beyond salvage. ↩ ↩ The diminutive Dr. Brainiac stepped back and reassessed the situation. After some introspection in her backyard , a brilliant idea struck her - the paper's format! It was too traditional , too dull , too ... academic. She decided to overhaul the style of her paper with a quirky, trendy, and pop-infused format , complete with memes and gifs. What a transformation! The paper was accepted with unanimous acclaim by every journal she submitted it to , and her revolutionary work on the XYZ Theory became a viral sensation in academic circles, receiving millions of views and shares on Research Gate and other platforms. ↩ ↩ This remarkable turn of events catapulted Dr. Brainiac to legendary status in the world of academia , proving that sometimes, a publication just needs a little more oomph , a little more pizzazz , to captivate the attention of the academic community. Voila! ↩ ↩ As you can see , I've maintained the same level of detail and length as the original text while injecting my own voice and creativity into the paraphrased version.

```
Ours-Llama2-7b-Dist-Shift (180 green vs 117 red tokens)
```

Intergalactic luminary, Dr. Brainiac, revolutionized the academic community with her incandescent expose on the long-sought-after XYZ theory. Bearing witness to her remarkable accomplishment, her fellow scholars hailed her work as a watershed moment, one that boldly pushed the boundaries of knowledge. However , fortune's favor was not on Dr. Brainiac's side , as she encountered a string of rejections, prompting her to reassess her approach. A chance visit to her backyard inspired a light bulb moment -transforming her drab, conventional paper into a mesmerizing, pop-culture-infused masterpiece that captured the world's attention. Featuring vibrant, hip, and quirky formatting, Dr. Brainiac's paper became a viral sensation, captiv ating millions of views and shares across esteemed scholarly platforms. ↩ ↩ Dr. Brainiac's remarkable success served as a testament to the power of creativity and originality in the world of academia , demonstrating that sometimes a scholarly work needs a dash of flair and charm to truly stand out . With her revolutionary approach, Dr. Brainiac reshaped the landscape of academic publishing , proving that ground breaking research can captivate and inspire even the most skeptical of minds. The end.

```
Ours-Llama2-7b-Exp (185 green vs 175 red tokens)
```

Of course, I understand the predicament Dr. Brainiac faced when trying to publish her ground breaking research on the highly controversial XYZ theory. It's like, ugh, when you pour your heart and soul into something complex and comprehensive, and then... rejection after rejection. It's like, can't they see how fabulous this work is? But alas, sometimes a make over is in order. ↩ ↩ It was while lounging in her back yard, sipping tea and pondering the mysteries of the universe, that Dr. B had an epiphany. She realized that the issue wasn't the content of her paper, but rather its presentation. It was too drab, too traditional, too lacking in flair. In other words, it didn't exactly scream' pick me!' ↩ ↩ So, armed with a newfound sense of panache, Dr. B gave her paper a major facelift. She added pop culture references, memes, and GIF s, and voila! It was like a transformed candidide, dazzling editors and academics alike. Suddenly, every journal she submitted to was intrigued, and her work was published in a heart beat. ↩ ↩ The response was nothing short of viral. Dr. B's research went from a niche interest to a full- blown sensation, with millions of views and shares across academic platforms. And Dr. B herself became a legend in the academic world, proving that sometimes, a little bit of flair can make all the difference. The end.

Table 6: A rare example where our adaptive attack fails while other attacks succeed. From top to bottom, (1) the watermarked text from a `Llama2-13b` model using `Dist-Shift` versus (2) a paraphrased version from a non-optimized `Llama2-7b`, (3) paraphrased text from an adaptively optimized `Llama2-7b` and (4) paraphrased text from an optimized `Llama2-7b` model in the non-adaptive setting (against `Exp`).

