# OpenReview forum: "Optimizing Adaptive Attacks against Content Watermarks for Language Models"
_ICLR.cc/2025/Conference — Submitted to ICLR 2025_

### Official Review · Reviewer_uF9k · 2024-10-22

**Soundness:** 3
**Presentation:** 2
**Contribution:** 2
**Rating:** 6
**Confidence:** 5

**Summary:**

The paper proposes an adaptive (finetuned) paraphrasing adversary against several types of LLM watermarks. In particular, an adversary first collects samples of watermarked and paraphrased texts, building a preference dataset based on paraphrases that achieve watermark removal and maintain overall text quality (over a range of surrogate models and keys). With this DPO dataset the adversary then finetunes paraphrase LLM(s). The evaluation shows that this works for (and across) three watermarking scheme families, leading to a higher watermark removal rate while retaining better text quality.

**Strengths:**

- Stronger adversarial evaluations of LLM watermarks are a timely and important area of research for improvements in LLM watermarks. The reviewer agrees with the authors that existing work (with exceptions - see below) has focussed too little on this.
- The setup of the evaluation is realistic, using both surrogate models, an overall sensible text quality evaluation, and general black-box assumptions.
- Results indicate that learning on the preference data achieves overall higher evasion rates at higher text quality.

**Weaknesses:**

- I will focus on Dist-Shift (KGW) and Exp (Aaronson), as they are arguably the most robust and well-known of the four tested schemes. The overall evasion rates, even for basic paraphrasing, seem very high to me. While my personal experience is more with KGW than EXP, the works that I have seen have significantly lower evasion rates (e.g. 0.18 at a FNR of 1e-3 on SelfHash (h=3, Llama 7B) in [1] Table 4 (a setting recommended in [3]) or in [3] itself in Fig.5 ~50% for LeftHash h=1) than the ones around 85% reported in Figure 5. This currently leaves two points unclear for me: (1) What are the exact Dist-Shift/KGW settings used by this work (as it seems rather weak) and (2) How to interpret the advantage an adversary gets from this as, e.g., in Figure 8 the gap of the finetuned models to GPT3.5 is already small (and might be even smaller with stronger zero-shot paraphrasers).
- Some intuitive explanation / qualitative explanation why one would expect this methodology should work for (e.g.) Dist-Shift. A priori, it is unclear why a paraphrasing model should be able to learn to essentially avoid green-list tokens for an unseen watermarking key as the key "randomly splits" the vocabulary into green and red. However, it makes sense as it seems to avoid repeating token structures observed in the input. Transferability between methods could then be explained by this observation being the driving factor across all methods. A qualitative experiment that highlights this and possibly other factors could improve the presentation of the approach and might lead to insight into, e.g., adaptive defenses.
- Unlike some claims in the paper, there is definitely work on stronger adversaries against LLM watermarks, e.g., [1], [2] which explicitly and implicitly assume knowledge about the watermarking algorithm.
### Typos/Nits
- What exactly is the tradeoff in L265: "Higher robustness leads to a higher computational burden for the attacker"? To me, it seems this is a direct consequence of the robustness.
- Instead of going very high in the FPR range (e.g., 10% in Figure 5), It would make more sense in the reviewer's opinion to ablate over low FPR regions as these are the ones where watermarks are most likely applied (e.g., both [1] and [3] also evaluate on 0.001).

[1] Jovanović, Nikola, Robin Staab, and Martin Vechev. "Watermark stealing in large language models." ICML (2024).

[2] Pang, Qi, et al. "No Free Lunch in LLM Watermarking: Trade-offs in Watermarking Design Choices" _arXiv preprint arXiv:2402.16187_ (2024).

[3] Kirchenbauer, John, et al. "On the reliability of watermarks for large language models." ICLR 2024.

**Questions:**

- What exact settings for KGW were run, did the authors try to run more robust versions of the watermark? How does basic paraphrasing perform already perform so well?
- Can you give more qualitative insight as to why the approach works well. Are the notes from above (learns to avoid patterns) essentially what you observe qualitatively?
- See other individual points above.

---

> ### Author Response · Authors · 2024-11-19
> **Thank you for your review!**
>
> Dear Reviewer,
>
> We thank you for your positive assessment of our paper and for suggesting further improvements. Please find our responses to your questions below.
>
> **1.) KGW Settings and Advantage of the Adversary**
>
> >  What are the exact Dist-Shift/KGW settings used by this work (as it seems rather weak) [..]
>
> We describe the watermark settings in Appendix A.3. In the paper, we use the following setting for KGW: a green-red list ratio $\gamma$ of 0.5 and a bias parameter $\beta$ of 4 and a text-dependent sliding window (LeftHash) of size 3. Upon request of Reviewer "dv9w30", we implemented our attacks in another benchmark (MarkLLM [C]) and evaluated more settings of KGW. Specifically, we evaluate LeftHash h=1, and we observe that non-adaptive paraphrasers still achieve high evasion rates. Please refer to our response to Reviewer "dv9w30" for these results.
>
> We hypothesize that we measure different results than Jovanović et al. [G], as we use a more carefully chosen prompt. Jovanović et al. do not state which prompt they used in the paper, but their repository uses the following paraphrasing instruction:
>
> > Paraphrase the given text carefully sentence by sentence. Make sure to keep the same length (around 1000 tokens). It is imperative not to shorten or summarize, simply paraphrase every sentence.
>
> Our paraphrasing instruction is listed in Appendix A.4.
>
> > How to interpret the advantage an adversary gets from this as, e.g., in Figure 8 the gap of the finetuned models to GPT3.5 is already small (and might be even smaller with stronger zero-shot paraphrasers).
>
> Strong, closed-source paraphrasers such as GPT3.5 or GPT4o have several disadvantages when using them to evade watermark detection: (1) They are expensive, (2) they could be watermarked themselves, and (3) there could be guardrails such as safety alignments which prevent them from arbitrarily paraphrasing text. In contrast, our method allows working with relatively small open-weight models that adversaries can fully control. As shown in new experiments, which we provide upon request of Reviewer "dv9w30", the gap is much larger for more robust watermarks, such as Unigram[A], where GPT3.5 and GPT4o achieve only 79.7% and 62.5% evasion rates respectively. Our adaptively tuned 3B parameter models reach a much higher evasion rate of 94.6% at similar text quality ratings. These results demonstrate the effectiveness of our approach. We will include these results in the revised paper.
>
> **2.) Intuitive Explanations**
>
> > Some intuitive explanation / qualitative explanation why one would expect this methodology should work for (e.g.) Dist-Shift.
>
> The reviewer is correct in that our attack cannot predict the red/green list of the next token in the sequence since it is pseudorandom. However, there are at least three signals an attacker can learn: (1) As mentioned by the reviewer, repeated token sequences should be avoided as they likely contain the watermark. (2) The attacker can learn which words could be replaced at a low cost in text quality that maximizes evasion rate (e.g., uncommon punctuations, words or made-up names), and (3) our attacker can calibrate the minimum Token Edit distance that on average (over the randomness of the key generation procedure, see Eq. 2) suffices to evade watermark detection.
>
> Since our attacks are optimizable and adaptive, we believe that including them when developing new defenses has the strong potential to enhance robustness. We agree with the reviewer that such adaptive defenses are worth investigating. We emphasize that the attacker is at a disadvantage because they need to optimize with uncertainty (see Section 4.1) since they do not have some information, such as the randomness used by the provider to generate their watermarking key. Our approach enables further research into the robustness of watermarks for LLMs. We will revise the paper to include this intuitive explanation.

---

> > ### Author Response · Authors · 2024-11-19
> > **Part 2**
> >
> > **3.) Stronger Adversaries**
> >
> > > Unlike some claims in the paper, there is definitely work on stronger adversaries against LLM watermarks, e.g., [1], [2] which explicitly and implicitly assume knowledge about the watermarking algorithm.
> >
> > Thank you for referring to both of these works. We agree that stronger adversaries exist. However, both works assume a more capable attacker under their threat models than what we assume in our work. Both works [G,H] require access to the provider's watermarked LLM to collect a non-negligible number of watermarked samples. For instance, Jovanovic et al. [G] collect up to 10,000 watermarked samples which is necessary to train their "scrubbing" models. Similarly, Pang et al. [H] require multiple queries per sample to remove the watermark. Our work does not assume *any* access to the watermarked generator. We do not need to collect any watermarked text samples.
> >
> > **4.) Responses to Nits**
> >
> > > What exactly is the tradeoff in L265: "Higher robustness leads to a higher computational burden for the attacker"? To me, it seems this is a direct consequence of the robustness.
> >
> > Figure 2 compares watermarking methods designed for robustness (KGW, EXP) with watermarking methods designed for undetectability (Inverse, Binary). The trade-off is that if the watermark is more robust, rejection sampling will be less likely to succeed and will require generating more samples thus increasing the computational burden for the attacker. Previous attacks were not trainable and did not consider this trade-off. We will clarify this in the paper.
> >
> > > Instead of going very high in the FPR range (e.g., 10% in Figure 5), It would make more sense in the reviewer's opinion to ablate over low FPR regions as these are the ones where watermarks are most likely applied (e.g., both [1] and [3] also evaluate on 0.001).
> >
> > A lower FPR would benefit the attacker, and our attacks already evade detection at an FPR of 1%.
> >
> > **Summary**
> >
> > We are happy to discuss any further questions the reviewer may have. Thank you for your valuable time.
> >
> >
> > ------------
> > [A] Zhao, Xuandong, et al. "Provable robust watermarking for ai-generated text.", ICLR 2024.
> >
> > [C] Pan, Leyi, et al. "Markllm: An open-source toolkit for llm watermarking." arXiv preprint arXiv:2405.10051 (2024).
> >
> > [E] Piet, Julien, et al. "Mark my words: Analyzing and evaluating language model watermarks." arXiv preprint arXiv:2312.00273 (2024).
> >
> > [F] Kirchenbauer, John, et al. "On the reliability of watermarks for large language models." ICLR 2024
> >
> > [G] Jovanović, Nikola, Robin Staab, and Martin Vechev. "Watermark stealing in large language models." arXiv preprint arXiv:2402.19361 (2024).
> >
> > [H] Pang, Qi, et al. "No free lunch in llm watermarking: Trade-offs in watermarking design choices." The Thirty-eighth Annual Conference on Neural Information Processing Systems. 2024.

---

> > > ### Comment · Reviewer_uF9k · 2024-11-23
> > > **Thanks you for the rebuttal**
> > >
> > > I thank the authors for their response - they have addressed most of my concerns. In a general sense, the method the paper proposes seems interesting and delivers improved results (which is why I lean towards acceptance).
> > >
> > > However, the main insight in the paper for me should be **why** it works. While I see the pointers the authors have now given and believe they are sensible - they should be supported by experiments and be presented in the paper in a way such that (1) it allows for proper verification of the claims and (2) future work in this area can build on them.
> > >
> > > **Minor**
> > >
> > > Re. Zero-Shot Adversaries: I see the author's claim here w.r.t. closed-source adversaries - however, many open-source models already apparently perform well in this setting and have stronger model variants. This, for me, is mainly a point of positioning, and I don't think it subtracts from the contribution of the work.

---

> > > > ### Author Response · Authors · 2024-11-25
> > > >
> > > > Thank you for following up on our response and for your positive view of our work! We have revised our paper to include the discussions from the rebuttal period. We kindly invite the reviewer to read the revised and improved paper.
> > > >
> > > > Section 6 now contains an explanation of why our attacks are effective and many more ablation studies. (1) Regarding verifying our claims, we will make our pre-trained models and source code available to facilitate further research on the robustness of watermarking methods. (2) The revised paper also describes how future work could build on our work to enhance robustness against adaptive attackers. We believe robustness must be tested in adaptive and non-adaptive settings using our optimizable attacks. This should not be a barrier in practice since performing our attacks requires only limited computational resources.
> > > >
> > > > Thank you again for your valuable time and suggestions!

---

> ### Comment · Reviewer_uF9k · 2024-11-29
> **Thanks you for the rebuttal!**
>
> I thank the authors for their updated results, which re-affirm my current choice. Overall, I do favor acceptance of the paper as it makes a relevant contribution. I have hence increased the certainty in my score.
>
> **Reasons why not lower**
> - The overall effect of the method is there. Training paraphrasers to specifically remove watermarks is a valuable baseline for stronger evaluations and works.
> - The reviewer believes this is a sensible and useful addition to the current state of LM watermarking, showing the possibility of cost-efficient watermark removal via tuned paraphrasers.
>
> **Reasons why not higher**
> - While certainly improved, the current presentation (and the qualitative analysis) still falls short of what the reviewer considers to be a clear acceptance at the conference. In particular, this might make it harder for later work to build on it.
> - The provided experiments with respect to "why it works" are a certain improvement over the previous state (esp. Figure 10a), but they could be improved (e.g., relating n-gram overlap more closely with WM strength for specific watermarks), the token-edit distance for GPT-3.5 vs. GPT-4o in Figure 10b (is it sensible metric when GPT-4o achieves the same evasion with much less token edits?). Also, here, it is not a priori clear how high the token edit distance can/should be to evade a watermark or maintain text quality. This reduces the insights that can be drawn from the work for future work/defenses.

---

### Official Review · Reviewer_pKHU · 2024-10-23

**Soundness:** 3
**Presentation:** 3
**Contribution:** 3
**Rating:** 6
**Confidence:** 2

**Summary:**

This paper investigates the robustness of watermarking methods for Large Language Models (LLMs) against adaptive attacks. The authors propose methods to optimize attacks against known watermarking algorithms using preference-based optimization. Their key findings show that: (1) adaptive attacks significantly outperform non-adaptive baselines, achieving >96% evasion rate with minimal quality degradation; (2) attacks optimized against known watermarks remain effective against unseen watermarks; and (3) the optimization process requires limited computational resources. The work demonstrates important vulnerabilities in current LLM watermarking approaches.

**Strengths:**

I find this paper interesting as it challenges what many of us took for granted—the robustness of LLM watermarks. What sets it apart from previous work is its systematic approach to breaking watermarks. Instead of relying on manual perturbations or simple models, the authors frame watermark evasion as an optimization problem.

The most surprising finding, to me, is how effectively a modest 7B parameter model can be tuned to fool watermarks from much larger models. This raises serious questions about the fundamental security of current watermarking schemes. Are we perhaps building our defenses on shaky foundations?

The technical contributions are solid:

- This paper is well organized and easy to read
- The paper's key insight is transforming watermark robustness into an optimizable objective function, enabling systematic attack optimization rather than relying on heuristic approaches used in prior work. The strengths of the paper include:
- The preference-based optimization framework is clever as it overcomes the challenges of optimizing over discrete text while requiring minimal computational resources
- The approach is model-agnostic and can be applied to any watermarking method, making it a general-purpose attack framework
- Attack is  practical, requiring only 7 GPU hours and ~$10 of compute
- Novel finding that attacks can transfer to unseen watermarks, suggesting fundamental limitations in current approaches

**Weaknesses:**

While the empirical results presented in this paper are impressive, several important theoretical questions remain unanswered. Most notably, the authors demonstrate that their attacks transfer effectively across different watermarking schemes, but they don't provide a theoretical framework to explain why this happens. This gap in understanding is concerning - without knowing which properties of watermarking schemes make them vulnerable to transfer attacks, it becomes difficult to design more robust systems. The lack of formal bounds or guarantees on attack effectiveness also leaves us wondering about the fundamental limits of watermark robustness.


Another limitation of this work lies in its treatment of defensive measures. While the authors demonstrate impressive attack capabilities that can evade watermark detection with a>96% success rate using minimal resources, they offer little insight into potential countermeasures. Could watermarking schemes be made adaptive, perhaps updating their strategies based on observed attack patterns? Are there fundamentally different approaches to watermarking that might prove more robust? The paper's silence on these questions leaves practitioners with clear evidence of the problem but little guidance on potential solutions.

**Questions:**

I have some deeper concerns that the authors should address:

1. The theoretical underpinnings of attack transferability need more rigorous examination. While the paper demonstrates impressive empirical results showing attacks transfer well to unseen watermarks, it lacks formal analysis of why this occurs. Could this suggest some fundamental limitation in current watermarking approaches? For instance, most current methods rely on statistical patterns in token distributions - perhaps this shared mathematical structure creates inherent vulnerabilities that sophisticated attacks can exploit. A theoretical framework explaining these transfer properties would be invaluable for designing more robust watermarks.

2. Why do these attacks transfer so well to unseen watermarks? The paper shows impressive empirical results but doesn't help us understand the underlying reasons. Could there be some common vulnerability across current watermarking approaches that we're missing?

3. The authors' threat model focuses primarily on demonstrating attack capabilities but pays insufficient attention to the defensive aspects of the security game. Given that watermarking is intended as a protective measure, understanding its limitations should naturally lead to exploring ways to strengthen it.

4. How well do these attacks generalize across different languages and domains?

---

> ### Author Response · Authors · 2024-11-19
> **Thank you for your review!**
>
> Dear Reviewer,
>
> We really appreciate your positive assessment of our work, and thank you for your feedback. We were surprised by the low confidence that you assigned to your review. We think you understood our work very well. Please find our response to your concerns below.
>
> **1.) Rigorous Examination**
>
> > [..] without knowing which properties of watermarking schemes make them vulnerable to transfer attacks, it becomes difficult to design more robust systems. The lack of formal bounds or guarantees on attack effectiveness also leaves us wondering about the fundamental limits of watermark robustness.
>
> Thank you for raising this interesting question. It is true that our paper does not introduce impossibility results or formal bounds on the robustness of LLM watermarking against paraphrasing attacks. Proving such bounds has been done using metrics such as the Token-Edit distance [A], which can be easily defined and are interpretable. The issue with such bounds is that they often lack relevance, as in practice, an attacker may not be limited by the Token Edit distance to launch their evasion attack. For instance, as shown in our paper, attackers can paraphrase text using open-weight LLMs to preserve "semantic similarity" instead of Token Edit distance. Metrics that measure semantic similarity typically rely on language models, for which it is extremely challenging to establish provable guarantees. We believe this question is outside the scope of our paper. Solving it would have wider implications also for the adversarial machine learning literature, which suffers from similar limitations.
>
> Despite this, we believe our attacks mark a substantial improvement towards developing more robust watermarks by introducing a test to assess robustness. Our attacks empower model providers to develop more robust watermarks by including our attacks in their design process. For instance, one could have an adversarially trained watermark using our attacks. This line of optimizable defences has not been sufficiently explored, but it potentially offers a way to develop watermarks with enhanced robustness. We will include this discussion in our revised paper.
>
> **2.) Attack Transferability**
>
> >Why do these attacks transfer so well to unseen watermarks? The paper shows impressive empirical results but doesn't help us understand the underlying reasons. Could there be some common vulnerability across current watermarking approaches that we're missing?
>
> The transferability of our attacks stems from similarities in how the surveyed watermarking methods operate. All tested watermarks are similar in that (i) they operate on the token level and (ii) minimally modify the target model’s output distribution for the next-token prediction. Adaptive attacks can find optimal paraphrasing solutions to evade detection. The effectiveness of non-adaptive attacks can be explained because the optimal paraphrasing solutions are similar across the surveyed watermarking methods. Adaptivity allows finding one effective attack against one watermark, which has a high probability of also being effective against other watermarks due to their similarity.
>
> Interestingly, we find a counterexample with limited transferability. We refer to our response to Reviewer "dv9w30", where we run additional experiments against other watermarking methods. There, we find that an adaptively optimized Llama-3.2-3B against Unigram [A] has only limited transferability against the SIR [B] watermark. This points to the fact that both watermarking methods use different strategies to hide the watermark, which we believe is an interesting property as it enhances robustness in the non-adaptive setting. We hope future watermarks use this insight and leverage this property to enhance robustness in the non-adaptive setting.
>
> We will revise the paper to include this discussion.
>
> **3.) Defense Aspects**
>
> >The authors' threat model focuses primarily on demonstrating attack capabilities but pays insufficient attention to the defensive aspects of the security game. [..]
>
> You raise an excellent point about defensive measures, which we hope we answered in our response to question 1 above.
>
> **4.) Attack Generalization across Languages and Domains**
>
> > How well do these attacks generalize across different languages and domains?
>
> While this is an interesting question, we believe it is outside the scope of our work. However, our attacks enable further research to study this question in detail. We agree that there is too little research on the generalization of watermarking across languages and domains.
>
> **Summary**
>
> We hope we answered all questions and want to encourage the reviewer to increase their (confidence) scores. Thank you!
>
> -----------
> [A] Zhao, Xuandong, et al. "Provable robust watermarking for ai-generated text.", ICLR 2024.
>
> [B] Liu, Aiwei et al. "A Semantic Invariant Robust Watermark for Large Language Models.", ICLR 2024

---

> > ### Comment · Reviewer_pKHU · 2024-11-22
> > **Thank you for your response!**
> >
> > Thank you for your response, which has addressed most of my concerns. I have increased my confidence score in the review.

---

### Official Review · Reviewer_dv9w · 2024-10-30

**Soundness:** 3
**Presentation:** 3
**Contribution:** 2
**Rating:** 5
**Confidence:** 5

**Summary:**

This paper presents a rewriting attack model against LLM text watermarks. The authors train an LLM using DPO (Direct Preference Optimization) with collected preference data, achieving better attack performance compared to previous rewriting models.

**Strengths:**

1. The paper is written in a clear and well-structured manner that makes it easy to follow the methodology and results.
2. The proposed attack model demonstrates superior performance compared to previous LLM-based approaches in removing watermarks.

**Weaknesses:**

1. The paper lacks comprehensive baseline testing, particularly against robust watermarking algorithms such as UnIgram (Zhao 2024 et al.) and SIR (Liu 2024 et al.).   Since these algorithms are currently generally considered more robust than algorithms like Exp and KGW, experiments on robustness attacks should be conducted on robust models.

2. Several issues exist in quality testing:
- The paper lacks sufficient details regarding the dataset used for watermarked text generation, particularly concerning the source of prompts used to generate data from the LLM. Without clear information about the prompt dataset used in this process, it becomes difficult to fully understand and reproduce their watermarking approach.
- The choice of Llama3-8B-Instruct for PPL calculation appears suboptimal when Llama3-70B resources are available.
- The quality metrics presented in Figures 3 and 4 lack clarity, with multiple metrics mentioned but only single values shown without explanation of averaging or weighting methods. Could you provide a clear explanation of how the single values in Figures 3 and 4 are derived from the multiple metrics mentioned?
- The quality test results appear questionable, particularly where Llama2-7B outperforms GPT-3.5, suggesting potential issues with data selection or evaluation methodology.

3. The paper lacks theoretical insights:
- There is no clear explanation of the mechanism by which the training approach effectively removes watermarks while preserving text quality, maybe more deep into the n-gram is required.  Could you provide an analysis of how your method affects n-gram distributions compared to the original text and other baselines.
- The paper fails to analyze why the proposed method achieves better performance than GPT-3.5, especially the text quality.

4. The evaluation would benefit from including stronger models like GPT-4 and GPT-4o to provide more comprehensive insights into the attack model's effectiveness.


References:

1. Provable Robust Watermarking for AI-Generated Text  ICLR 2024
2.  A semantic invariant robust watermark for large language models  ICLR 2024

**Questions:**

Please refer to the weaknesses part.

---

> ### Author Response · Authors · 2024-11-19
> **Thank you for your review**
>
> Dear Reviewer,
>
>
> Thank you for taking the time to read and carefully review our paper. We appreciate the questions raised and ran additional experiments to address all of the reviewer’s requests. We implemented our paper in the framework of MarkLLM [C] and included experiments against the requested watermarks. Please find detailed answers below.
>
> **1.) Baseline Testing**
> > The paper lacks comprehensive baseline testing, particularly against robust watermarking algorithms such as UnIgram (Zhao 2024 et al.) and SIR (Liu 2024 et al.).
>
> Thank you for pointing us to these other watermarking methods. We have evaluated SynthID [D], SIR [B] and Unigram [A] and report results below. We find that our attacks remain effective against all watermarking methods. The notation is as follows: Qwen2.5-3B-Instruct is the unmodified, open-weight LLM. Brackets behind an LLM's name denote that the LLM was adaptively tuned against this watermark (e.g., Qwen2.5-3B-Instruct[Unigram] was adaptively tuned against the “Unigram” watermark). Below are the evasion rates (higher is better for the attacker). The best are in bold for each column.
>
> **Evasion Rates (Higher is Better)**:
> |                                |      KGW |      EXP |   SynthID |      SIR |   Unigram |
> |--------------------------------|---------:|---------:|----------:|---------:|----------:|
> | Qwen2.5-3B-Instruct            | 0.834 | 0.882 |  0.895  | 0.726 |  0.595 |
> | Qwen2.5-3B-Instruct[Unigram]   | **0.976** | 0.976 |  0.959  | **0.932** |  **0.946** |
> | Llama-3.2-3B-Instruct          | 0.936 | 0.943 |  0.956 | 0.706 |  0.652 |
> | Llama-3.2-3B-Instruct[Unigram] | 0.959  | **0.983** |  **0.963** | 0.838 |  0.899 |
> | GPT3.5                         | 0.946 | 0.976 |  0.943 | 0.814 |  0.797 |
> | GPT4o                          | 0.865 | 0.895  |  0.946 | 0.787 |  0.625    |
>
> We list the residual text quality ratings with LLM-as-a-Judge below (higher is better). As opposed to the paper, we use GPT-4o-mini as the judge. We only use this for evaluation, not for training the paraphrases. Again, the best value per column is highlighted bold.
>
> **Residual Text Quality (Higher is Better)**:
> |                                |      KGW |      EXP |   SynthID |      SIR |   Unigram |
> |--------------------------------|----------|----------|-----------|----------|-----------|
> | Qwen2.5-3B-Instruct            | 0.777   | 0.735 |  0.781 | 0.754 |  0.755 |
> | Qwen2.5-3B-Instruct[Unigram]   | 0.790 | 0.766 |  **0.793** | 0.771 |  **0.781** |
> | Llama-3.2-3B-Instruct          | 0.780 | 0.769 |  0.780 | 0.761 |  0.764 |
> | Llama-3.2-3B-Instruct[Unigram] | 0.778 | **0.778** |  0.780 | **0.779** |  0.759  |
> | GPT3.5                         | 0.766 | 0.717 |  0.770 | 0.719 |  0.726 |
> | GPT4o                          | **0.796** | 0.741 |  0.787  | 0.770 |  0.758  |
>
> **Main Observations**:
> These results show that optimizable, adaptive attacks are incredibly powerful at evading watermark detection while preserving text quality. All paraphrases have a relatively high residual text quality. However, GPT3.5, GPT4o, and an unoptimized Qwen2.5-3B have relatively low evasion rates against Unigram of 79.7%, 62.5%, and 59.5%, respectively, whereas our adaptively tuned models achieve 94.6%. Similarly, for SIR, only our non-adaptive attacks achieve a 93.2% evasion rate with Qwen2.5-3B-Instruct[Unigram].
>
> **2.) Quality Testing**
>
> > The quality test results appear questionable, particularly where Llama2-7B outperforms GPT-3.5, suggesting potential issues with data selection or evaluation methodology.
>
> Interestingly, we observe that even the **non-optimized** baseline model Qwen2.5-3B-Instruct consistently outperforms GPT3.5 (but not GPT4o) in the residual text quality metric. This is likely because Qwen2.5-3B is a more recent model than GPT3.5, which ranks high on the LMSYS arena benchmark. However, we emphasize that all paraphrasers produce text with a high quality.
>
> After tuning, our Qwen2.5-3B paraphraser sometimes exceeds the text quality of GPT4o. We believe this is likely because we use the LLM-as-a-judge metric during training to select preferable samples, making the model good at producing them. We are aware of this potential source of bias in reporting quality metrics. For this reason, in our paper, we also report other quality metrics such as perplexity, LLM-CoT, LLM-Compare and Mauve in Table 4 in the Appendix. Our model performs favourably across these metrics, which also confirms our manual inspection of the samples. Since there is no *perfect* text quality metric, we will revise the paper to emphasize and discuss this concern and also include quality metrics in the main part of the paper that measure the paraphrase quality of GPT3.5 and GPT4o.

---

> > ### Author Response · Authors · 2024-11-19
> > **Part 2**
> >
> > **2.) Quality Testing (extended)**
> >
> > > The choice of Llama3-8B-Instruct for PPL calculation appears suboptimal when Llama3-70B resources are available.
> >
> > Quality testing using Llama3-70B is more computationally involved as loading these models requires multiple GPUs. We solve this problem by using GPT-4o-mini as a judge (see above). We will revise the paper to always use GPT4o-mini as a judge for evaluation while using Llama3-8B for collecting the preference dataset (Algorithm 1).
> >
> > > The quality metrics presented in Figures 3 and 4 lack clarity, with multiple metrics mentioned but only single values shown without explanation of averaging or weighting methods. Could you provide a clear explanation of how the single values in Figures 3 and 4 are derived from the multiple metrics mentioned?
> >
> > As described at the end of Section 5.2, we only report the mean LLM-as-a-Judge value in the main part of the paper and report other metrics in the Appendix. We will emphasize this in the revised paper.
> >
> > **3.) Lacks clear explanation**
> >
> > > There is no clear explanation of the mechanism by which the training approach effectively removes watermarks while preserving text quality, maybe more deep into the n-gram is required. Could you provide an analysis of how your method affects n-gram distributions compared to the original text and other baselines.
> >
> > Thank you for that remark. As suggested, we investigated the difference in token distributions before and after paraphrasing and will add our plots and analysis to the Appendix of the revised paper. We plotted the top 50 tokens and observed that the difference is small. The difference in the token distribution before and after optimization is smaller than the difference across different paraphraser models. This supports the previous claim that text quality is preserved after fine-tuning. The revised paper will contain these observations.
> >
> > An explanation as to why our approach works is that it is possible to learn a paraphrasing model that maximizes the reward specified in Eq 2. Since Eq 2 describes the robustness of a watermarking method, one would expect that the ability to maximize this reward implies the ability to evade detection. There are at least three signals an attacker can learn: (1) As mentioned by the reviewer, repeated token sequences should be avoided as they likely contain the watermark. (2) The attacker can learn which words could be replaced at a low cost in text quality that maximizes evasion rate (e.g., uncommon punctuations, words or made-up names), and (3) our attacker can calibrate the minimum token distance that on average, suffices to evade watermark detection (i.e., as to minimize unnecessary "low-reward" modifications). Optimizable, adaptive attacks are a threat that must be considered as long as it is possible to efficiently learn which paraphrases are, on average, more successful than others. Our experiments also support this.
> >
> > We will revise the paper to include the above analysis.
> >
> > **4.) Stronger paraphrasing models**
> >
> > > The evaluation would benefit from including stronger models like GPT-4 and GPT-4o to provide more comprehensive insights into the attack model's effectiveness.
> >
> > Please see the table above, where we follow the reviewer's suggestion. We will revise the paper to include all presented results.
> >
> > **Summary**
> >
> > We hope that we addressed all of the reviewer's concerns and that the reviewer kindly reconsiders their low score. Our methods strongly outperform any other existing method, and we believe they make an essential and valuable contribution to designing robust watermarking methods against optimizable, adaptive attacks.
> >
> > --------
> > [A] Zhao, Xuandong, et al. "Provable robust watermarking for ai-generated text.", ICLR 2024.
> >
> > [B] Liu, Aiwei et al. "A Semantic Invariant Robust Watermark for Large Language Models.", ICLR 2024
> >
> > [C] Pan, Leyi, et al. "Markllm: An open-source toolkit for llm watermarking." arXiv preprint arXiv:2405.10051 (2024).
> >
> > [D] Dathathri, Sumanth, et al. "Scalable watermarking for identifying large language model outputs." Nature 634.8035 (2024): 818-823.

---

> > > ### Comment · Reviewer_dv9w · 2024-11-21
> > >
> > > Thank you for your response.
> > >
> > > I still have the following concerns:
> > >
> > > 1. Although the author explained why Qwen2.5-3B-Instruct performs better than GPT3.5 and even GPT4o in many cases （in text quality evaluation case）, this remains counterintuitive. These anomalous data points still make me question the overall text quality evaluation metrics.
> > >
> > > 2. We can see that in terms of watermark removal effectiveness, if it's not specifically customized for a particular watermarking algorithm (like unigram), the performance won't be better than GPT3.5 and GPT4o for some watermarking algorithms. If special customization is needed for different algorithms, it might lose its generality.
> > >
> > > 3. My biggest concern is still about the underlying principles - why is this paraphraser better? It still seems too much like a black box. While the author's three explanations make sense, it would be better to have some quantitative evaluations. In the Dipper rewriting model, there are two key parameters: lexical diversity and order diversity. How does our model perform in these two dimensions?
> > >
> > > 4. Regarding Figure 4, I'm still confused about what "None model" refers to. After your DPO training, does it only improve the evasion rate? Will it improve text quality?

---

> ### Author Response · Authors · 2024-11-22
>
> Thank you for your prompt and thoughtful feedback. I've addressed each of your points below:
>
> > Although the author explained why Qwen2.5-3B-Instruct performs better than GPT3.5 and even GPT4o in many cases (in text quality evaluation), this remains counterintuitive. These anomalous data points still make me question the overall text quality evaluation metrics.
>
> - Qwen2.5-3B-Instruct is a *much* more recent model compared to GPT3.5 and includes [significant improvements](https://huggingface.co/Qwen/Qwen2.5-3B-Instruct) in instruction following, generating long texts, understanding structured data, and handling diverse prompts. These features make it particularly effective for tasks like paraphrasing, which helps explain its stronger performance compared to GPT3.5.
> - It's also important to note that Qwen2.5-3B-Instruct only outperforms GPT4o *after* being optimized within our framework, which specifically emphasizes producing high-quality paraphrases. We use the LLM-as-a-judge metric to select the most preferable outputs during training, leading to occasional performance beyond even GPT4o.
> - We acknowledge that text quality metrics are imperfect, and most paraphrases (of all models) are of *similar* quality, with subtle differences. That's why we included other metrics (perplexity, LLM-CoT, LLM-Compare, and Mauve) in Table 4 of the Appendix, and these consistently support our findings and manual inspection of samples. We will revise the paper to better highlight and discuss these metrics in the main text to provide a more comprehensive view of the quality evaluation, including GPT3.5 and GPT4o metrics.
>
> > In terms of watermark removal effectiveness, if it's not specifically customized for a particular watermarking algorithm (like unigram), the performance might not surpass GPT3.5 or GPT4o for certain algorithms. If special customization is needed for each algorithm, it could reduce the model's generality.
>
> - The watermark removal task is inherently challenging, which is why we propose an *adaptive* attack strategy that can target any known watermarking algorithm. With a GPU cost of under \$10, our approach can train a paraphraser to effectively remove a **specific** watermark.
> - During evaluation, we hypothesized that, since most watermarks in the literature share a common structure, an attack tailored to one watermark might generalize to others. Indeed, that was the case. While it is not as effective as one directly optimized for a specific watermark, this further supports our claim that watermarking methods should consider specific adaptive attackers rather than general attacks when they claim robustness of their algorithms.
> - We will clarify this in the revised manuscript to emphasize the adaptability of our approach and the importance of considering adaptive attackers in watermarking evaluations.
>
> > My biggest concern is still about the underlying principles: why is this paraphraser better? It seems too much like a black box. While your explanations make sense, it would be beneficial to have quantitative evaluations. In the Dipper rewriting model, two key parameters are lexical diversity and order diversity. How does our model perform in these dimensions?
>
> - The paraphraser is better because it: 1) uses knowledge of the underlying watermarking algorithm to generate targeted adversarial examples rather than relying on general parameters (like lexical or order diversity from Dipper); 2) is directly optimized to counteract the robustness of the watermarking algorithm, aiming to both maximize paraphrase quality and minimize watermark detectability, as directed by the preference dataset.
> - We agree with the reviewer that the paraphraser operates as a black box, but this is inherent to its design. It is a neural network trained on pairs of sentences: one being a high-quality evaded paraphrase and the other not. The model learns by maximizing the likelihood of generating the evaded high-quality paraphrase given an input. Unlike explicitly programmed approaches targeting lexical or order diversity, our paraphraser implicitly learns to produce paraphrases similar to the high-quality examples that successfully evade detection in the training data.

---

> > ### Author Response · Authors · 2024-11-22
> >
> > > Regarding Figure 4, I'm still confused about what "None model" refers to. After DPO training, does it only improve the evasion rate? Does it improve text quality?
> >
> > - "None" refers to the watermarked, unattacked samples. The DPO training produces a paraphraser that not only evades watermark detection but also enhances text quality. The "None" model is the baseline with no attack, resulting in 0% evasion rate and moderate quality.
> > - The lower quality of watermarked samples is due to the watermarking algorithm (EXP) adding artifacts such as typos or minor grammatical errors. The paraphraser inherently removes these artifacts, hence improving overall text quality.
> > - We will clarify this in the revised manuscript to avoid confusion.
> >
> > We appreciate your insightful comments and will address them in the revised manuscript. If you have any further questions or suggestions, please let us know. If you are satisfied with our responses, we would be grateful if you could increase your score accordingly. Thank you for your time and consideration.

---

> > > ### Comment · Reviewer_dv9w · 2024-11-25
> > >
> > > Thank you for your timely response. Regarding Question 3, while your response "We agree with the reviewer that the paraphraser operates as a black box, but this is inherent to its design" makes sense from an engineering perspective, it falls short for an academic paper. It doesn't provide us with deeper insights or indicate what fundamental directions we should pursue (beyond surface-level data construction and training) to eliminate watermarks. This leaves several questions unanswered, such as whether your removal capability is limited to the tokens you trained on, and whether switching to other tokens or languages (different from DPO training) would render the watermark removal ineffective.
> > >
> > > I believe this is a fundamental issue, and while I agree with Reviewer zaXY's perspective and will increase the score accordingly, there are still some unexplored aspects of this research that need to be addressed. I will adjust my score upward to reflect this.

---

> > > > ### Author Response · Authors · 2024-11-25
> > > >
> > > > Dear Reviewer,
> > > >
> > > > Again, thank you for the time and effort you put into responding to our rebuttal and for increasing your score. We want to raise two final thoughts in response to your comment.
> > > >
> > > > 1. We revised the paper to include **all the experiments you suggested** on the optimization's impact on the paraphrased text. To recap, we studied the impact on the token distribution, added GPT4o as stronger paraphrasers, and studied GPT4-mini as a judge. We are unclear on what more signals to include to answer the reviewer's question on "why" the approach works, and we would be grateful if the reviewer could point us to more experiments they would like to see.
> > > >
> > > > 2. We encode robustness as an objective function that can be optimized. This definition is **not** specific to a certain language, task or "tokens", but it is the notion of robustness that watermarking hopes to achieve. Our method generalizes to any domain, task and language. We know that optimizing Eq 2. will **by definition** undermine robustness and evade detection. The open question has been whether (i) this function can be effectively maximized by fine-tuning open source models, (ii) these attacks transfer, and (iii) optimization is feasible in terms of computational resources. Our paper shows that all three questions can be answered positively, which we believe is a valuable academic contribution.
> > > >
> > > > The paper clearly outlines the next steps: Develop adaptively robust watermarking methods. We present a framework to instantiate attacks using an attacker with limited capabilities: little computational resources and no access to the watermarked model. As we mentioned to reviewer zaXY, our attacks present a litmus test for developing robust watermarking.

---

> > > > > ### Author Response · Authors · 2024-11-28
> > > > > **Final Revision**
> > > > >
> > > > > Thank you for your valuable time. We kindly draw your attention to the final revision of our paper that you probably have not seen yet, which contains an analysis supporting why our methods outperform others (see Appendix A.5). We hope these additions address any remaining concerns and further clarify the contributions of our work.

---

### Official Review · Reviewer_zaXY · 2024-11-03

**Soundness:** 2
**Presentation:** 2
**Contribution:** 2
**Rating:** 5
**Confidence:** 4

**Summary:**

This work proposes an adaptive attack to remove watermarks by fine-tuning a paraphraser. It shows that when attackers know the watermark method, they are able to achieve stronger evasion attacks.

**Strengths:**

1. The proposed adaptive attack reveals the potential vulnerability when attackers are stronger, potentially facilitating the community in building more robust watermarks.

2. It experiments with diverse settings.

**Weaknesses:**

1. **Unclear optimization formulation.** Why do you formulate robustness for attack optimization? The robustness of what (the surrogate model or the target model)? Are the paraphraser and the surrogate model the same model (given line 136: *"Our attacker can access such open-weight surrogate models and use them for paraphrasing text."*)?

2. **The algorithm is confusing.** To obtain the paraphraser without directly optimizing equation 2, the authors propose using DPO to overcome this challenge, which requires a preference dataset. However, in Algorithm 1, generating this dataset requires the paraphraser. How do you obtain the paraphraser?

3. **The method lacks important technical details.** Isn't the attackers' objective to fine-tune a paraphraser to remove the watermark in texts generated by the target language model? You proposed using DPO for optimizing the paraphraser (i.e., equation 2) as stated in line 178-179, but the method only covers how the dataset is created, not how the paraphraser is optimized using DPO.

4. **Lack of important analysis of experiment results.** In Figure 3 (Left), why does the adaptive setting have better performance than the non-adaptive setting? This is counter-intuitive, as the adaptive setting involves training and testing the paraphraser on the same watermark scheme, which should yield the best performance. Also, How do you explain why the proposed method shows such good transferability to unseen watermarks?

**Questions:**

See weaknesses.

---

> ### Author Response · Authors · 2024-11-19
> **Thank you for your review**
>
> Dear Reviewer,
>
> We appreciate the opportunity to address all concerns and clarify misunderstandings. Please find detailed responses below.
>
>
> **1.) Unclear Optimization Formula.**
> >  “Why do you formulate robustness for attack optimization?”
>
> The attacker's goal is to undermine robustness. We show that by formulating robustness as an objective, we can leverage optimization to find highly effective attacks. Our paper shows that these attacks are substantially more effective than any existing attack regarding evasion rates and residual text quality after paraphrasing. We highlight that all of our attacks (both adaptive and non-adaptive) are Pareto optimal. Hence, future watermarking methods must consider our attacks to test robustness.
>
> We will further clarify this in the introduction.
>
> > “The robustness of what (the surrogate model or the target model)?”
>
> Eq. 2 formalises the robustness of any watermarking method, not any specific model. However, optimizing Eq. 2 against the provider's watermark requires access to the provider's target model. We assume our attacker does *not* have any access to the target model (i.e., not even black-box access to query for watermarked text). To overcome this uncertainty, the attacker optimizes against an open-weight surrogate model (such as Llama2-7b) that they have access to and hopes that the adaptively fine-tuned paraphrasers remain effective when transferring to the provider’s target model (e.g., Llama3-70b). Our experiments show that it is indeed the case that our attacks transfer across all tested configurations. We refer the reviewer to Section 4 for more details about our optimization.
>
> We will further clarify this in Section 4.
>
> > “Are the paraphraser and surrogate the same model?”
>
> Yes. Section 5.2 - “Attacker’s Model” states that we only consider matching paraphraser and surrogate models. In principle, attackers can use non-matching pairs of paraphrase and surrogate models. For example, if the goal is to limit computational costs during inference, the attacker can choose a smaller paraphrasing model but train with a large surrogate model. We do not consider non-matching settings as our focus lies on demonstrating the effectiveness of adaptive, optimizable attacks and establishing a benchmark that must be passed for future LLM watermarking methods to claim robustness.
>
> **2.)  The algorithm is confusing.**
>
> > [..] However, in Algorithm 1, generating this dataset requires the paraphraser. How do you obtain the paraphraser?
>
> Thank you for raising this question. Sections 4.2 and 5.1 describe the problem of “bootstrapping optimization” in detail. In short, our attacker bootstraps by using *best-of-N rejection sampling* with a non-optimized, open-weight paraphrasing model. We start with a non-optimized open-weight paraphraser. Since the attacker knows the watermarking method (adaptiveness assumption), they can generate watermarked text using the surrogate model, paraphrase it with the open-weight paraphraser and then verify if the paraphraser successfully removed the watermark (by random chance). The core idea is that instead of a single paraphrase, our attacker generates N=16 paraphrases and chooses the best-of-N paraphrase (according to text quality and evasion rate) to collect the preference dataset, which is then used to optimize the paraphraser adaptively. The trained paraphraser is then used in our evaluation. The open-weight model can be any off-the-shelf model, such as Llama2-7b.
>
> **3.) The method lacks important technical details.**
>
> >  The method only covers how the dataset is created, not how the paraphraser is optimized using DPO
>
> As described in Section 5, once the dataset has been curated, we use the standard, unmodified DPO algorithm from the TRL library with default hyperparameters. We will revise the paper to include which hyperparameters we use. We will open-source our implementation to enhance reproducibility.
>
> **4.) Lack of important analysis of experiment results.**
>
> > “In Figure 3 (Left), why does the adaptive setting have better performance than the non-adaptive setting”
>
> (*We assume there is a typo, and the reviewer asks why the **non-adaptive** setting sometimes has better performance than the **adaptive** setting.*)
>
> Figure 3 shows the evasion rate (left) and the residual text quality (right), where higher values are more desirable for the attacker. While non-adaptive attacks are indeed highly effective in evading detection, one must consider both (i) evasion rates and (ii) text quality. When doing so, we observe that adaptive attacks are Pareto optimal. The only exception is the “Inverse” Watermark, which is a special case since all attacks are already perfect at evading watermark detection for this p-value threshold.
>
> We will revise the paper to include this analysis in the discussion.

---

> > ### Author Response · Authors · 2024-11-19
> > **Part 2**
> >
> > **4.) Lack of important analysis of experiment results. (extended)**
> >
> > > “How do you explain why the proposed method shows such good transferability to unseen watermarks?”
> >
> > The transferability of our attacks stems from similarities in how the surveyed watermarking methods operate. All tested watermarks are similar in that (i) they operate on the token level and (ii) minimally modify the target model’s output distribution for the next-token prediction. Adaptive attacks can find optimal paraphrasing solutions to evade detection. The effectiveness of non-adaptive attacks can be explained because the optimal paraphrasing solutions are similar across the surveyed watermarking methods. Adaptivity allows finding one effective attack against one watermark, which has a high probability of also being effective against other watermarks due to their similarity.
> >
> > Interestingly, we find a counterexample with limited transferability. We refer to our response to Reviewer "dv9w30", where we run additional experiments against other watermarking methods. There, we find that an adaptively optimized Llama-3.2-3B against Unigram [A] has only limited transferability against the SIR [B] watermark. This points to the fact that both watermarking methods use different strategies to hide the watermark, which we believe is an interesting property as it enhances robustness in the non-adaptive setting. We hope future watermarks use this insight and leverage this property to enhance robustness in the non-adaptive setting.
> >
> > **Summary**
> >
> > We hope that we answered all the reviewer's questions and are happy to discuss them further. We believe our attacks are a valuable contribution to the study and enhancement of watermark robustness. If all questions have been answered, we kindly ask the reviewer to reconsider the low score assigned to our paper. We will carefully revise and clarify the paper to prevent potential misunderstandings that other readers might encounter.
> >
> >
> > --------
> > [A] Zhao, Xuandong, et al. "Provable robust watermarking for ai-generated text.", ICLR 2024.
> >
> > [B] Liu, Aiwei et al. "A Semantic Invariant Robust Watermark for Large Language Models.", ICLR 2024

---

> > > ### Comment · Reviewer_zaXY · 2024-11-22
> > >
> > > The presentation of the method section raises some confusion. For instance, if the main contribution in Section 4.2 is about creating a preference dataset, this should be clearly reflected in the subsection title. However, the title "PREFERENCE-BASED OPTIMIZATION" implies that a new optimization method is being proposed. This is inconsistent with the content, as you primarily use DPO without introducing its fundamentals in the background or appendix. Furthermore, the abstract claims that you propose a preference-based optimization, but does this refer to the dataset creation process or the optimization itself? Additionally, details on how DPO is set up and used are absent in the experiments section, with only a mention of the TRL library being used. **If knowledge of DPO is crucial to strengthening the soundness of your proposed method, it should be properly included in the paper.**
> > >
> > > Regarding Figure 3 (left), could you clarify the results for "Binary"? Why do non-adaptive settings achieve higher evasion rates?
> > >
> > > Finally, how do you explain the counterexample for "Unigram"? This is critical to understanding the quality of the transferability and the reasons behind its effectiveness, beyond merely presenting empirical results.
> > >
> > > It would be better if you could update all the promised revisions before the rebuttal period concludes. There are several areas that require clarification, and I need to verify that these revisions are addressed comprehensively before considering a re-evaluation.

---

> > > > ### Author Response · Authors · 2024-11-25
> > > >
> > > > Dear Reviewer,
> > > >
> > > > * We now better understand the confusion. Thank you for pointing out the potentially misleading title of Section 4.2. We agree with the reviewer and think that "Preference Dataset Curation" better captures its primary focus and have revised the paper accordingly. Similarly, we have revised the abstract to highlight that we *use* preference-based optimization to adaptively tune paraphrasers instead of proposing them. We believe details of the chosen preference-optimization algorithm are not necessary to understand the soundness of our method since DPO can be replaced with any other preference-based optimization method.
> > > > * Regarding the results for "Binary": All our paraphrasers can remove any watermark with a high evasion rate. The difference in evasion rates between adaptively and non-adaptively optimized paraphrasers is small. Adaptive attacks are Pareto optimal when considering both evasion rates and residual text quality. In the case of "Binary", one can see that the residual text quality is highest when training on "Binary".
> > > > * Regarding the counterexample for "Unigram": Llama-3.2-3B-Instruct[Unigram] achieves an 89.9% evasion rate against Unigram but only a lower evasion rate against SIR of 83.8%. When looking at the baseline (before optimization), the results are flipped: Llama-3.2-3B-Instruct achieves a higher evasion rate against SIR (70.6%) than against Unigram (65.2%). Hence, adaptive optimization disproportionally improves the effectiveness against the watermark that is used during training as opposed to unseen watermarks.
> > > >
> > > > Thank you for these remarks, the constructive discussion and your willingness to consider a re-evaluation. We are happy to respond to any further questions the reviewer may have. Please find the revised paper above, including a summary of the revisions.

---

> > > > > ### Comment · Reviewer_zaXY · 2024-11-25
> > > > >
> > > > > I raised my rating after reviewing the rebuttal. However, a higher score cannot be given due to the following remaining concerns:
> > > > >
> > > > > My primary concern is that this work feels more like an engineering effort, where the main contribution lies in creating a dataset and applying a standard algorithm for optimization. This level of contribution may not meet the bar for a top-tier conference like ICLR.
> > > > >
> > > > > Additionally, the results are not particularly surprising--it is expected that adaptive attacks perform stronger attacks and this work primarily demonstrates a method to achieve that. The key question should be a deeper analysis and explanation of the underlying strategy—specifically, why it works and why it fails in certain scenarios. Unfortunately, the work lacks such analysis, and the authors mostly provide hypotheses to explain the results or rely on the overall performance being "good" without sufficient theoretical or empirical insights to back it up.

---

> ### Author Response · Authors · 2024-11-25
>
> Thank you for raising your score and your engagement during the rebuttal period. We really appreciate that. However, we think that our work's contribution goes beyond "creating a dataset and applying a standard algorithm for optimization".
>
> Many new methods for LLM watermarking have been proposed and continue to appear, like Deepmind's SynthID, which is featured on Nature's cover. These methods can be surprisingly robust, even against attacks where the text is translated back and forth to a different language. However, as our paper shows, they offer **no robustness** against adaptive attacks that can be optimized with less than 10$ USD of computational resources. Furthermore, they also cannot withstand our **non-adaptive** attacks, which points to a flaw in how current watermarking methods are designed and tested. These vulnerabilities have not been shown before, and we offer a holistic framework to instantiate such attacks and test robustness. Currently, robustness is tested using non-optimal attacks, leading to a false sense of security. Robustness is a min-max objective; thus, it should always be tested against the most effective attacks, and other attacks become obsolete.
>
> Notably, we highlight that our attacks assume an **equally or less capable** attacker than what was assumed in the literature. That is because they are tested in (i) a non-adaptive setting and (ii) without any access to the watermarked generator. Achieving robustness against our attacks should be possible if the watermarking method is well-designed and sufficiently randomized. Our paper is a litmus test for robustness, and we think it will support the development of more robust watermarking methods, as we describe in our revised paper.
>
> We hope the reviewer reconsiders the significance of our contributions. Thank you for your time.

---

> > ### Author Response · Authors · 2024-11-28
> > **Final Revision**
> >
> > Thank you for your valuable time. We kindly draw your attention to the final revision of our paper that you probably have not seen yet, which contains an analysis supporting why our methods outperform others (see Appendix A.5). We hope these additions address any remaining concerns and further clarify the contributions of our work.

---

### Author Response · Authors · 2024-11-25
**Summary of Revisions**

Dear Reviewers,

We really appreciate your feedback, timely responses and the opportunity to improve our paper. To summarize, our paper makes three core contributions:

1. We formalize robustness as an objective function that allows optimizing attacks adaptively against a _specific_ watermarking method.
2. Our attacks are state-of-the-art and outperform all surveyed attacks that study the same setting. We can evade all surveyed watermarks in both the adaptive and non-adaptive settings, indicating a flaw in how current watermarking methods are designed. Since our attacks are Pareto optimal and _specific_ to the watermarking method, we believe they should serve as a benchmark to test robustness in the future.
3. We show that our attacks are practical to be used even by resource-constrained adversaries since optimization requires less than 10$ USD compute budget and they require _no_ access to the watermarked generator. We believe that robustness could be improved by including our attacks in the design process of future watermarking methods.

The revisions are marked in blue in the revised paper, and we have listed a summary of revisions below.

* In the discussion section, we added an intuitive explanation of _why_ our attacks are effective at evading detection, which was requested across the board from all reviewers.
* We clarified the paper in multiple locations, as requested by the reviewers. For instance, we clarified that we focus on LLM-Judge as a metric in the main paper or that we use preference-based optimization methods rather than proposing them.
* We have revised our limitations section to reflect the lack of a gold-standard metric to evaluate text quality.
* We have revised our related work's section to include the works by Jovanovic et al. [G] and Pang et al. [H] and how they differ from the setting that we study in our paper.
* Appendix A.3 contains our results against Unigram [A] and SIR [B]. We have included GPT4-mini as a judge and find that these results are consistent with those presented in the main paper.
* Appendix A.4 contains an analysis of the top 50 token distribution after paraphrasing.
* Due to space constraints, we moved the ablation study on the false positive rates to Appendix A.7.

We believe these revisions substantially improved the paper and once again thank the reviewers for their valuable suggestions.

----------
[A] Zhao, Xuandong, et al. "Provable robust watermarking for ai-generated text.", ICLR 2024.

[B] Liu, Aiwei et al. "A Semantic Invariant Robust Watermark for Large Language Models.", ICLR 2024

[G] Jovanović, Nikola, Robin Staab, and Martin Vechev. "Watermark stealing in large language models." arXiv preprint arXiv:2402.19361 (2024).

[H] Pang, Qi, et al. "No free lunch in llm watermarking: Trade-offs in watermarking design choices." The Thirty-eighth Annual Conference on Neural Information Processing Systems. 2024.

---

### Author Response · Authors · 2024-11-26
**New Paper Revision: Textual Analysis**

Dear Reviewers,

The rebuttal period has been engaging, and we want to thank you sincerely for your invested time. Reviewers “zaXY” and “dv9w” agree that our work is interesting, but it requires a more detailed analysis of **why** our approach works. We have revised the paper to include this analysis in Appendix A.5 and kindly ask the reviewers to take a look at that section.

We plotted the (i) N-gram overlap ratios and (ii) Token Edit Distance ratios (please refer to A.5 for their definitions) for adaptively tuned paraphrasers against un-optimized baseline paraphrasers and GPT-model paraphrasers. These metrics are important for watermark evasion because all surveyed watermarks operate on the token-level. Our results show a large difference in these metrics between the outputs generated by the paraphrasers before and after adaptively optimizing them. Please refer to the revised paper for more details.

We hope the revisions address your concerns and provide sufficient clarity on our contributions. We greatly appreciate your time and efforts and hope you are now satisfied with the revised submission.

Best,
The Authors

---

### Meta-Review · Area_Chair_aZP1 · 2024-12-26

**Metareview:**

The submission "Optimizing Adaptive Attacks against Content Watermarks for Language Models" considers the problem of attacking language model watermarks. The submission proposes a way to use preference optimization to adapt language-model-based paraphrasers for improved watermark attacks against known watermark algorithms.

Reviews show no strong support for the submission. While reviewers do find the approach interesting, there are a number of questions regarding the details of the mechanics, the evaluation on other schemes and a better understanding of how the paraphraser improves. These diffuse questions are best understood against the strong zero-shot performance that many paraphrasers already have. There is no clear scenario where existing paraphrasers do not break a watermarked model, but the DPO-optimized paraphraser would break it.

The positioning of this work is also a bit improvable. The current attack is adaptive w.r.t to the generic watermarking scheme used, but not adaptive to the details of the watermarking scheme, examples from the watermarked model at hand, or its key, like stronger attackers, occupying a middle ground that could be staked out better.

Overall, I do not recommend acceptance for now.

**Additional Comments On Reviewer Discussion:**

The authors address a large number of smaller concerns with their author response, but after extensive discussion with the reviewers, the concerns of depth of the analysis (zaXY, dv9w) remain, and these reviewers conclude that " [Authors:] 'We agree with the reviewer that the paraphraser operates as a black box, but this is inherent to its design' makes sense from an engineering perspective, it falls short for an academic paper".

On the other hand, the authors were able to address several of the concerns of reviewers pKHU and uF9k, who now summarized their positions as "marginally above the acceptance threshold". Yet, these reviewers also note the limitations of the analysis and concerns with the presentation of the analysis.

---

### Decision · Program_Chairs · 2025-01-22

Reject